# Enhancing Affine Maximizer Auctions with Correlation-Aware Payment

## Abstract

Affine Maximizer Auctions (AMAs), a generalized mechanism family from VCG, are widely used in automated mechanism design due to their inherent dominant-strategy incentive compatibility (DSIC) and individual rationality (IR). However, as the payment form is fixed, AMA's expressiveness is restricted, especially in distributions where bidders' valuations are correlated. In this paper, we propose Correlation-Aware AMA (CA-AMA), a novel framework that augments AMA with a new correlation-aware payment. We show that any CA-AMA preserves the DSIC property and formalize finding optimal CA-AMA as a constraint optimization problem subject to the IR constraint. Then, we theoretically characterize scenarios where classic AMAs can perform arbitrarily poorly compared to the optimal revenue, while the CA-AMA can reach the optimal revenue. For optimizing CA-AMA, we design a tailored loss function with a two-stage training algorithm. We derive that the target function's continuity and the generalization bound on the degree of deviation from strict IR. Finally, extensive experiments showcase that our algorithm can find an approximate optimal CA-AMA in various distributions with improved revenue and a low degree of violation of IR.

## 1 Introduction

Differentiable economics [9, 16, 40, 43] has recently attracted significant attention within automated mechanism design. By leveraging advanced neural network architectures and gradient-based optimization algorithms, these approaches construct auctions demonstrating superior empirical performance. In revenue-maximizing auction design, existing methods are broadly categorized into two classes: (1) *characterization-free* methods, which directly employ neural networks to approximate auction mechanisms [13, 16, 25, 34, 37], and (2) *characterization-based* methods, which optimize within structured mechanism families possessing well-defined economic properties [9, 14, 15, 40, 43]. Among the latter, Affine Maximizer Auctions (AMAs), a family of mechanisms extended from Vickery-Clarke-Groves (VCG) [27, 42], are particularly notable for inherently guaranteeing dominant-strategy incentive compatibility (DSIC), individual rationality (IR), and preventing over-allocation. Recent work on optimizing AMAs has demonstrated strong empirical performance and balanced computational efficiency [9, 14, 15].

However, prior AMA-based methods have primarily focused on evaluations under bidder-independent distributions, where the limitations in expressiveness of VCG-style payment rules may not be fully apparent. In certain bidder-correlated settings, this VCG-style payment rule exhibits a critical constraint: a bidder's payment can only be a non-decreasing function of other bidders' valuations. This inherent limitation significantly reduces their payment flexibility compared to characterization-free [16] or menu-based mechanisms [43]. For instance, consider a single-item auction with two bidders where valuations are perfectly negatively correlated ($v_1 = 1 - v_2$) and each marginal valuation

is drawn uniformly from $[0, 1]$. Here, the optimal mechanism extracts full surplus by setting a reserve price of $1 - \min\{v_1, v_2\}$. This implies that when $v_1 > 0.5$, bidder 1 is allocated the item and pays $1 - v_2$. Yet, for any AMA, the payment when bidder 1 wins must be non-decreasing in $v_2$, thus rendering it incapable of expressing such a simple optimal mechanism.

Motivated by this limitation, we aim to enhance AMA's expressiveness in bidder-correlated settings while preserving its structural advantages and optimization efficiency. *Existing research on bidder-correlated auctions has largely concentrated on theoretical designs, predominantly for single-item settings.* The seminal Crémer-McLean auction [7, 8] established conditions under which DSIC and Bayesian IR mechanisms can extract full surplus. Subsequent studies have analyzed the computational complexity [6, 12, 33] and sample complexity [1, 19, 44] of optimal auctions under specific correlated priors, while others have investigated the robustness of existing mechanisms (e.g., second-price auctions) to correlation [5, 21, 45]. Closest to our work are [24] and [17]: Huo et al. [24] proposed a score-based payment rule trained via max-min neural networks, approximating optimal revenue in *single-item auctions*; The result by Feldman and Lavi [17] implies limitations of classic AMAs compared to optimal *interim IR mechanism*. Our results show that AMA can perform badly even compared to the optimal *ex-post IR mechanism*.

In this paper, we introduce the Correlation-Aware Affine Maximizer Auction (CA-AMA), which incorporates an additional correlation-aware payment term, $p_i^{\text{Cor}}$, for each bidder. Since $p_i^{\text{Cor}}$ is independent of bidder $i$'s bid, CA-AMA inherently maintains the DSIC property, and we formalize the problem of identifying the optimal CA-AMA as an optimization problem subject to IR constraints. Theoretically, we demonstrate that in single-item auctions under certain distributions, CA-AMA can achieve optimal revenue where classic AMAs perform arbitrarily poorly. We then derive a tailored loss function and a two-stage training algorithm for optimizing CA-AMA. The algorithm's feasibility is supported by the continuity of the target function and a generalization bound on the degree of IR violation. Finally, we conduct extensive experiments across various distributions in single-item and multi-item auctions. The results demonstrate our algorithm's effectiveness in finding an approximately IR CA-AMA and achieving significantly improved revenue compared to classic AMAs.

The remainder of this paper is organized as follows: Section 2 introduces preliminaries. Section 3 demonstrates the limitations of classic AMAs and proposes the CA-AMA framework. Section 4 details the optimization of CA-AMA. Experimental results are presented in Section 5, and Section 6 concludes the paper. More details about related work are in Section A.

## 2 Preliminary

We consider the sealed-bid auction with $n$ bidders $[n] = \{1, 2, \ldots, n\}$ and $m$ items $[m] = \{1, 2, \ldots, m\}$. Each bidder $i$ has a private valuation on all item combinations, denoted by $\boldsymbol{v}_i = (v_{is})_{s \subseteq [m]}$, where $v_{is}$ is the bidder's valuation of an item combination $s \subseteq [m]$. We mainly consider the *additive* valuation, *i.e.*, $v_{is} = \sum_{j \in s} v_{ij}$ for all $i \in [n]$ and $s \subseteq [m]$. So a bidder's valuation is expressed by $\boldsymbol{v}_i = (v_{ij})_{j \in [m]}$.

A valuation profile $V = (\boldsymbol{v}_1, \boldsymbol{v}_2, \ldots, \boldsymbol{v}_n)$ is a collection of all bidders' valuations. We assume that $V$ has an underlying distribution $\mathcal{F}$ and the support is bounded, $\text{supp}(F) \subseteq [0, 1]^{n \times m}$. In an auction, each bidder $i$ reports a bid $\boldsymbol{b}_i$, which does not necessarily equal its real valuation $\boldsymbol{v}_i$. The auctioneer does not know the true valuation profile $V$ nor the distribution $\mathcal{F}$ but can observe the bidding profile $B = (\boldsymbol{b}_1, \boldsymbol{b}_2, \ldots, \boldsymbol{b}_n)$. We use $V_{-i} = (\boldsymbol{v}_1, \ldots, \boldsymbol{v}_{i-1}, \boldsymbol{v}_{i+1}, \ldots, \boldsymbol{v}_n)$ to represent the valuation profile except for bidder $i$, and $B_{-i}$ with the similar meaning. The marginal distribution is represented by $\mathcal{F}_i(V_{-i})$ for bidder $i$'s valuation. When the bidders are *independent*, this marginal distribution does not depend on $V_{-i}$, which means that $\mathcal{F}_i(V_{-i}) \equiv \mathcal{F}_i$ for any $V_{-i}$. When the bidders' valuation distributions are *correlated*, such a relationship does not hold.

### 2.1 Revenue-Maximizing Auction Design

An auction mechanism $(g, p)$ consists of an allocation rule $g$ and a payment rule $p$. For a given bidding profile $B$, $g(B) \subseteq [0, 1]^{n \times m}$ is the allocation matrix. The allocation rule has to satisfies that $\sum_{i=1}^{n} g(B)_{ij} \leq 1$ for any $j \in [m]$. If the mechanism is *deterministic*, we further restrict that the allocation matrix $g(B)_{ij} \subseteq \{0, 1\}$ for all $i$ and $j$. The payment rule $p_i(B) \geq 0$ determines the

value the bidder $i$ has to pay. Following the literature [14, 16], we assume that the bidders are utility maximizers and have quasi-linear utility. For a mechanism $(g, p)$, the utility of bidder $i$ with true valuation $\boldsymbol{v}_i$ when the bid profile is $B$ can be written as $u_i(\boldsymbol{v}_i, B; g, p) := \boldsymbol{v}_i \cdot g(B)_i - p_i(B)$. If the mechanism $(g, p)$ which we are referring to does not raise ambiguity, we will use $u_i(\boldsymbol{v}_i, B)$ for simplicity.

The auction mechanism will be announced publicly at first so bidders can statically report their valuation to gain a higher utility. We consider the following properties from classic auction theory [32].

**Definition 2.1.** A mechanism $(g, p)$ satisfies *dominant-strategy incentive compatibility* (DSIC) if

$$u_i(\boldsymbol{v}_i, (\boldsymbol{v}_i, B_{-i})) \geq u_i(\boldsymbol{v}_i, (\boldsymbol{b}_i, B_{-i})), \quad \forall i, B_{-i}, \boldsymbol{v}_i, \boldsymbol{b}_i. \tag{DSIC}$$

**Definition 2.2.** A mechanism $(g, p)$ satisfies *individual rationality* (IR) if

$$u_i(\boldsymbol{v}_i, (\boldsymbol{v}_i, B_{-i})) \geq 0, \quad \forall i, B_{-i}, \boldsymbol{v}_i \in \text{supp}(\mathcal{F}_i(B_{-i})). \tag{IR}$$

Note that the definition is different from the *ex-post IR*, which requires $u_i(\boldsymbol{v}_i, (\boldsymbol{v}_i, B_{-i})) \geq 0$ for all $i$, $\boldsymbol{v}_i$ and $B_{-i}$. This is weaker than ex-post IR but stronger than ex-interim IR, as it requires the utility to be non-negative on each point $(\boldsymbol{v}_i, B_{-i})$ that can be realized by $\mathcal{F}$. As the mechanism we analyze in this paper always satisfies DSIC, it is reasonable to assume that all bidders will truthfully report and so that we can exclude some valuation profiles that will never be realized.

The optimal auction design is to find the revenue-maximizing DSIC and IR auction mechanism under a certain distribution $\mathcal{F}$, which can be formulated as the following optimization problem.

$$\max_{g, p} \quad \text{REV}_{\mathcal{F}} := \mathbb{E}_{V \sim \mathcal{F}} \sum_{i=1}^{n} p_i(V) \tag{OPT}$$
$$\text{s.t.} \quad \text{Mechanism } (g, p) \text{ satisfies DSIC and IR.}$$

## 2.2 Affine Maximizer Auctions

AMAs is a family of auction mechanisms generalized from the VCG [27, 42] auction. An AMA can be parameterized by $(\mathcal{A}, \boldsymbol{w}, \boldsymbol{\lambda})$. $\mathcal{A} = \{A_1, \cdots, A_S\}$ is a set of $S$ distinct candidate allocations, $w_i$ is the weight for bidder $i$ and $\lambda_k$ is the boost for allocation $A_k$. A deterministic AMA refers to the AMA whose parameter $\mathcal{A}$ is fixed by all possible deterministic allocations, and so that $S = (n+1)^m$ (each item can be allocated to any of the $n+1$ bidders).

Formally, with the parameter set as $(\mathcal{A}, \boldsymbol{w}, \boldsymbol{\lambda})$, denote $\text{asw}(k; V) := \sum_{i=1}^{n} w_i(\boldsymbol{v}_i \cdot (A_k)_i) + \lambda_k$ the affine social welfare for $k$-th allocation under valuation profile $V$ and $\text{asw}_{-i}(k; V) = \text{asw}(k; V) - w_i(\boldsymbol{v}_i \cdot (A_k)_i)$, the allocation and payment rule can be written as

$$g^{\text{AMA}}(V; \mathcal{A}, \boldsymbol{w}, \boldsymbol{\lambda}) = A_{k^*} : \quad k^* = \arg\max_{k \in [S]} \text{asw}(k; V),$$
$$p_i^{\text{AMA}}(V; \mathcal{A}, \boldsymbol{w}, \boldsymbol{\lambda}) = \frac{1}{w_i} \left( \max_{k \in [S]} \text{asw}_{-i}(k; V) - \text{asw}_{-i}(k^*; V) \right). \tag{AMA}$$

As AMA satisfies DSIC and IR regardless of the chosen parameters [38, 39], the problem of finding the revenue-maximizing AMA with a fixed size of $\mathcal{A}$, $|\mathcal{A}| = S$, can be formulated as an unconstrained optimization.

$$\max_{\mathcal{A}:|\mathcal{A}|=S, \boldsymbol{w}, \boldsymbol{\lambda}} \quad \text{REV}_{\mathcal{F}}^{\text{S-AMA}} := \mathbb{E}_{V \sim \mathcal{F}} \sum_{i=1}^{n} p_i^{\text{AMA}}(V; \mathcal{A}, \boldsymbol{w}, \boldsymbol{\lambda}). \tag{AMA-OPT}$$

Specifically, *we denote $REV_{\mathcal{F}}^{D\text{-}AMA}$ the optimal revenue when fixing $\mathcal{A}$ to be the set of all deterministic allocations.* Recent work on AMA has its advantage of interpretability and strong performance in theory and empirical [28] shows that AMA is "approximately universal" under certain distributions, and recent AMA-based work [9, 14, 15, 39] attain considerable empirical performance when combined with machine learning approaches, even compared with those approximate DSIC auctions.

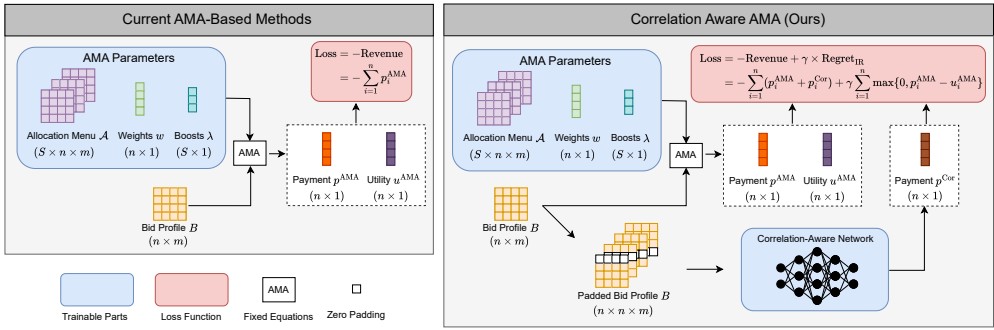

Figure 1: The comparison between the optimization for classic Affine Maximizer Auctions (AMAs) and our proposed Correlation Aware AMA (CA-AMA). In classic AMA-based methods [9, 14, 15, 39], we only optimize the AMA parameters to improve the revenue. To enhance AMA's performance under bidder-correlated distributions, we introduce a correlation-aware payment $p^{\text{Cor}}$ and hence add a $\text{Regret}_{\text{IR}}$ term in our loss function.

## 3 Correlation-Aware Affine Maximizer Auctions

This section begins by presenting a bidder-correlated single-item scenario where classic AMAs fail to achieve optimal revenue. We then define Correlation-Aware AMA (CA-AMA), a modification that introduces a correlation-aware payment term to enhance AMA's expressiveness while preserving the desirable property of DSIC. The problem of finding the optimal CA-AMA is subsequently formulated as an optimization problem constrained by IR. Finally, we provide a theoretical comparison of the revenue achievable by optimal CA-AMA and classic AMA in single-item auctions.

### 3.1 AMA Fails in Certain Correlated Distributions

We begin by analyzing a potential shortcoming of AMA-OPT. In current AMA-based methods [9, 14, 15, 39], AMA parameters $(\mathcal{A}, \boldsymbol{w}, \boldsymbol{\lambda})$ are determined during training and remain fixed at test time to ensure DSIC. Consequently, this static nature prevents the mechanism from further utilizing information from a specific input bidding profile $B$ during evaluation. Specifically, if bidders' valuations are linearly correlated, one bidder's valuation $\boldsymbol{v}_i$ can be inferred from the valuations of others, $V_{-i}$. To illustrate this deficiency, we construct an asymmetric correlated distribution $\mathcal{F}$ where the optimal AMA's revenue can be an arbitrarily small fraction of the optimal revenue.

**Proposition 3.1.** *In single-item auctions, for any number of bidders $n$ and any $\epsilon > 0$, there exists a distribution $\mathcal{F}$ such that $REV_{\mathcal{F}}^{D\text{-}AMA} \leq \epsilon \cdot REV_{\mathcal{F}}$. Furthermore, $REV_{\mathcal{F}}^{S\text{-}AMA} < REV_{\mathcal{F}}$ for any menu size $S$.*

The constructed distribution features a "dominant" bidder whose valuation is consistently the highest and is sampled from an equal revenue distribution. The valuations of other bidders are negatively linearly correlated with the dominant bidder's valuation, allowing the dominant bidder's exact valuation to be inferred from theirs. Under this distribution, a strict DSIC and IR mechanism can set a reserve price equal to the dominant bidder's valuation and hence extract full surplus. However, we show that any deterministic AMA can perform arbitrarily poorly, and even any randomized AMA fails to extract the full surplus. The underlying reason is that in allocation regions where the dominant bidder (say, bidder 1) does not receive the item, revenue is upper-bounded by the (low) valuations of other bidders. Conversely, in any region $[v, v']$ where the item is allocated to bidder 1, the AMA payment structure indicates that bidder 1's payment is non-decreasing in $V_{-1}$ and thus (due to the negative correlation) non-increasing in $v_1$. Consequently, the payment in this region is at most $v$, leading to sub-optimal revenue. This implies that even an optimally learned AMA will exhibit weak performance under such correlations.

### 3.2 Correlation-Aware Payment

Motivated by this failure case of classic AMAs, we propose a modification to address correlated valuation distributions. Specifically, we introduce an additional payment term for each bidder $i$,

$p_i^{\text{Cor}}(V_{-i})$, which depends solely on the valuations of other bidders, $V_{-i}$. Formally, the CA-AMA mechanism is defined as:

$$g^{\text{CA}}(V; \mathcal{A}, \boldsymbol{w}, \boldsymbol{\lambda}) = g^{\text{AMA}}(V; \mathcal{A}, \boldsymbol{w}, \boldsymbol{\lambda}),$$

$$p_i^{\text{CA}}(V; \mathcal{A}, \boldsymbol{w}, \boldsymbol{\lambda}, p^{\text{Cor}}) = p_i^{\text{AMA}}(V; \mathcal{A}, \boldsymbol{w}, \boldsymbol{\lambda}) + p_i^{\text{Cor}}(V_{-i}).$$

For any bidder $i$, since $p_i^{\text{Cor}}(V_{-i})$ depends only on other bidders' valuations, it acts as a constant from bidder $i$'s perspective when determining their optimal bid. Thus, the optimal bidding strategy remains unchanged from that in a classic AMA. Therefore, CA-AMA inherits the DSIC property from AMA.

**Proposition 3.2.** *For any $\mathcal{A}$, $\boldsymbol{w}$, $\boldsymbol{\lambda}$ and correlation-aware function $p^{\text{Cor}}$, the CA-AMA mechanism $(g^{CA}, p^{CA})$ satisfies DSIC.*

However, IR can be violated if $p_i^{\text{Cor}}(V_{-i})$ is set inappropriately high. Considering this, we formulate the problem of finding the optimal CA-AMA as an IR-constrained optimization problem:

$$\max_{\mathcal{A}:|\mathcal{A}|=S, \boldsymbol{w}, \boldsymbol{\lambda}, p^{\text{Cor}}} \quad \text{REV}_{\mathcal{F}}^{\text{S-CA}} := \mathbb{E}_{V \sim \mathcal{F}} \left[ \sum_{i=1}^{n} p_i^{\text{CA}}(V; \mathcal{A}, \boldsymbol{w}, \boldsymbol{\lambda}, p^{\text{Cor}}) \right] \quad \text{(CA-AMA-OPT)}$$

$$\text{s.t.} \quad \text{The mechanism } (g^{\text{CA}}, p^{\text{CA}}) \text{ satisfies IR.}$$

Similar to the notation for AMA, we denote $\text{REV}_{\mathcal{F}}^{\text{D-CA}}$ to be the optimal revenue obtained by CA-AMA with $\mathcal{A}$ fixed to be the set of all deterministic allocations. To highlight the importance of this formulation for correlated distributions, we analyze the relationship between the optimal revenues from AMA and CA-AMA. Our analysis primarily focuses on single-item auctions; we will also discuss the challenges in extending these theoretical guarantees to multi-item settings. The empirical performance of CA-AMA in multi-item auctions is demonstrated in Section 5.

Clearly, for any distribution $\mathcal{F}$, $\text{REV}_{\mathcal{F}}^{\text{CA}} \geq \text{REV}_{\mathcal{F}}^{\text{AMA}}$, since setting $p_i^{\text{Cor}}(V_{-i}) = 0$ for all $i$ allows CA-AMA to replicate any classic AMA. We then present cases where this relationship can be further characterized.

**Theorem 3.3.** *In single-item auctions, for any number of bidders $n$:*

- *If $\mathcal{F}$ is bidder-independent, then $REV_{\mathcal{F}}^{D\text{-}CA} = REV_{\mathcal{F}}^{D\text{-}AMA}$.*

- *For any $\epsilon > 0$, there exists a distribution $\mathcal{F}$ such that $REV_{\mathcal{F}}^{D\text{-}AMA} \leq \epsilon \cdot REV_{\mathcal{F}}$, while $REV_{\mathcal{F}}^{D\text{-}CA} = REV_{\mathcal{F}}$. Furthermore, $REV_{\mathcal{F}}^{S\text{-}AMA} < REV_{\mathcal{F}}$ for any menu size $S$.*

This result indicates that introducing the $p_i^{\text{Cor}}(V_{-i})$ term offers no benefit over classic AMAs in bidder-independent single-item auctions when considering deterministic mechanisms. The second part of the theorem utilizes the same constructed distribution as in Proposition 3.1. Under such correlated distributions, CA-AMA demonstrates significantly greater expressiveness than classic AMAs, achieving optimal revenue where AMAs fail. While this theorem pertains to single-item auctions, we *conjecture that similar results hold for multi-item auctions*. Proving this for multi-item auctions is challenging due to several factors: firstly, the optimal revenue in multi-item settings is often unknown, and characterizing the optimal AMA itself is difficult. Secondly, in multi-item auctions, the allocation of one item can be interdependent with others; for instance, an item might be reserved if bidders' valuations for other items are low, affecting overall allocation decisions. Therefore, we primarily validate the performance of CA-AMA in multi-item settings empirically in Section 5.

So far, we have introduced the CA-AMA framework, formulated its optimization problem, and theoretically analyzed its potential for revenue improvement over classic AMAs. The subsequent section will propose a data-driven algorithm for optimizing CA-AMA.

# 4 Optimization of CA-AMA

This section details the optimization procedure for finding the optimal CA-AMA. Within a data-driven framework, we first design a loss function tailored to CA-AMA-OPT. A two-stage training algorithm is proposed to optimize both the AMA parameters and the correlation-aware payments $p^{\text{Cor}}$. Furthermore, we establish the continuity of the optimal $p^{\text{Cor}}$ under mild assumptions and demonstrate that the generalization error for IR violation, i.e., the gap between training and test set performance, is bounded.

## 4.1 Loss Function Design

Analogous to definitions of regret on DSIC [16] and over-allocation [43], we define the $\text{Regret}_{\text{IR}}$ for a single data point $V$. This metric quantifies the extent of IR violation:

$$\text{Regret}_{\text{IR}}(g, p, V) := \sum_{i=1}^{n} \max\{0, -u_i(v_i, V; g, p)\}.$$

A mechanism satisfies IR if and only if $\text{Regret}_{\text{IR}}(g, p, V) = 0$ for all $V \in \text{supp}(\mathcal{F})$. To address the optimization problem CA-AMA-OPT, we design a loss function incorporating both the standard AMA payment $p_i^{\text{AMA}}$ and the correlation-aware term $p_i^{\text{Cor}}$. Given a dataset $D = \{V^{(1)}, V^{(2)}, \dots, V^{(K)}\}$ consisting of $K$ samples, the empirical loss is:

$$\mathcal{L}(\mathcal{A}, \boldsymbol{w}, \boldsymbol{\lambda}, \{p_i^{\text{Cor}}\}_{i=1}^{n}) := \sum_{k=1}^{K} \left[ -\text{Revenue}(V^{(k)}) + \gamma \cdot \text{Regret}_{\text{IR}}(V^{(k)}) \right]$$

$$= \sum_{k=1}^{K} \left( \sum_{i=1}^{n} - \left[ p_i^{\text{AMA}}(V^{(k)}) + p_i^{\text{Cor}}(V_{-i}^{(k)}) \right] + \gamma \sum_{i=1}^{n} \max \left\{ 0, p_i^{\text{Cor}}(V_{-i}^{(k)}) - u_i^{\text{AMA}}(V^{(k)}) \right\} \right).$$

$$\tag{Loss}$$

The loss function comprises the negative total revenue from the batch, derived from both $p^{\text{AMA}}$ and $p^{\text{Cor}}$, and the term penalizes IR violations. Note that a bidder's utility in CA-AMA, $u_i^{\text{CA}}(V)$, is their utility under the classic AMA minus the additional correlation-aware payment: $u_i^{\text{CA}}(V) = u_i^{\text{AMA}}(V) - p_i^{\text{Cor}}(V_{-i})$ for all $i \in [n]$. The hyperparameter $\gamma$ balances revenue maximization against IR satisfaction and is updated during training. Following Ivanov et al. [25], given a target regret for IR $R_{\text{target}}$, we estimate the batch $\text{Regret}_{\text{IR}}$ $R(D) = \frac{1}{K} \sum_{k=1}^{K} \text{Regret}_{\text{IR}}(V^{(k)})$ and update $\gamma$ iteratively:

$$\gamma_{t+1} = \text{clip}\left(\gamma_t + \gamma_{\Delta}\left(\log R(D) - \log R_{\text{target}}\right), 1, \bar{\gamma}\right),$$

where $\gamma_{\Delta}$ is the learning rate for $\gamma$, and $\bar{\gamma}$ is a predefined upper bound for $\gamma$.

## 4.2 Training

In our implementation, the AMA parameters $(\mathcal{A}, \boldsymbol{w}, \boldsymbol{\lambda})$ and the correlation-aware payments $p^{\text{Cor}}$ are determined by neural networks with parameters $\theta$ and $\phi$, respectively. To attain a mechanism with high revenue and low $\text{Regret}_{\text{IR}}$, we propose a two-stage optimization procedure: mutual training followed by post-training.

**Mutual Training.** In this stage, the parameters $\theta$ (for AMA components) and $\phi$ (for $p^{\text{Cor}}$) are jointly trained. Note that the Loss function is non-differentiable to the AMA parameters, and hence $\theta$, due to the argmax operation in the allocation rule. To enable gradient-based optimization, we follow [9, 14] to replace the argmax in the AMA allocation rule with a softmax approximation. This yields differentiable approximations for the AMA payments, $\hat{p}_i^{\text{AMA}}$, and utilities, $\hat{u}_i^{\text{AMA}}$, used in the loss function during this stage. The primary objective of mutual training is to find AMA parameters that are close to optimal for the combined objective. However, because the true AMA utility is approximated by $\hat{u}^{\text{AMA}}$, the actual regret of IR may not precisely meet the target $R_{\text{target}}$ after this stage. Therefore, a subsequent post-training stage is introduced to further refine $p^{\text{Cor}}$.

**Post-Training.** In this stage, the AMA parameters are frozen. Only the parameters $\phi$ are updated to fine-tune the correlation-aware payments $p^{\text{Cor}}$, aiming to maximize revenue while satisfying the target $R_{\text{target}}$. Since gradients of $\theta$ are not required, the exact AMA payments $p_i^{\text{AMA}}$ and utilities $u_i^{\text{AMA}}$ are used in the loss calculation for this stage. The rationale for fixing $\theta$ is that mutual training is assumed to have found a near-optimal configuration for the core AMA structure; post-training then performs a more precise adjustment of $p_i^{\text{Cor}}$. Furthermore, with fixed AMA components, optimizing $p_i^{\text{Cor}}$ becomes a more focused and potentially simpler problem than the joint optimization in the mutual training stage.

Detailed algorithmic descriptions for mutual training, post-training, and classic AMA optimization are provided in Appendix D.

### 4.3 Theoretical Characterizations

To conclude this section, we present theoretical results that support the validity and tractability of our optimization approach. Our theoretical analysis focuses on the novel aspects compared to classic AMA: the correlation-aware term $p^{\text{Cor}}$ and the $\text{Regret}_{\text{IR}}$ component of the loss.

**Continuity of Optimal $p^{\mathbf{Cor}}$.** For bidder $i$'s correlation-aware payment $p_i^{\text{Cor}}$, to maximize revenue subject to IR ($u_i^{\text{CA}} \geq 0$, which implies $u_i^{\text{AMA}}(V) - p_i^{\text{Cor}}(V_{-i}) \geq 0$), the largest such $p_i^{\text{Cor}}(V_{-i})$ is given by:

$$p_i^{\text{OPT-core}}(V_{-i}) := \inf_{\boldsymbol{v}_i \in \text{supp}(\mathcal{F}_i(V_{-i}))} u_i^{\text{AMA}}((\boldsymbol{v}_i, V_{-i}); \mathcal{A}, \boldsymbol{w}, \boldsymbol{\lambda}).$$

This means that to maximize revenue subject to IR, $p_i^{\text{Cor}}(V_{-i})$ should ideally be set to the minimum utility bidder $i$ would receive from the AMA mechanism. Intuitively, if $p_i^{\text{Cor}}(V_{-i})$ exceeds this value, IR is violated; if it is less, the revenue is sub-optimal. We then establish continuity properties for this $p_i^{\text{OPT-core}}$.

**Theorem 4.1.** *The target function $p_i^{OPT\text{-}core}$ is continuous with respect to the AMA parameters $\mathcal{A}$, $\boldsymbol{w}$, and $\boldsymbol{\lambda}$. Furthermore, assume that there exists a constant $C_H > 0$ such that for all $V_{-i}, V'_{-i}$, the Hausdorff distance $h(supp(\mathcal{F}_i(V_{-i})), supp(\mathcal{F}_i(V'_{-i}))) \leq C_H \|V_{-i} - V'_{-i}\|$, then $p_i^{OPT\text{-}core}$ is also continuous with respect to $V_{-i}$.*

This result demonstrates that the optimal $p_i^{\text{Cor}}(V_{-i})$ is continuous to both the AMA parameters and the input $V_{-i}$ under these mild assumptions. This continuity supports the feasibility of parameterizing $p_i^{\text{Cor}}$ with a neural network, which is a universal approximator for any continuous function [11, 23].

**Generalization Bound of $\text{Regret}_{\text{IR}}$.** We next provide a guarantee on the generalization of the IR regret term. This addresses the concern of whether a mechanism trained on a finite dataset will exhibit similarly low regret on unseen data drawn from the true underlying distribution $\mathcal{F}$. Specifically, we aim to show that the empirical $\text{Regret}_{\text{IR}}$, computed on the training set, is a reliable proxy for the true expected $\text{Regret}_{\text{IR}}$ under $\mathcal{F}$. Our analysis considers the post-training stage, where AMA parameters are fixed, and only $p^{\text{Cor}}$ is being learned. The following theorem bounds the difference between the empirical and expected $\text{Regret}_{\text{IR}}$.

**Theorem 4.2** (Informal version of theorem C.1)**.** *For each $i \in [n]$, let $p_i^{Cor}$ be the output of a 3-layer ReLU network whose weights have bounded spectral norms. Then, for any AMA parameters $(\mathcal{A}, \boldsymbol{w}, \boldsymbol{\lambda})$, distribution $\mathcal{F}$ and i.i.d. sample $D = \{V^{(1)}, \ldots, V^{(K)}\} \sim \mathcal{F}^K$, the following inequality holds uniformly over all such networks (i.e., all choices of parameters $\theta$):*

$$\sup_{\theta} \left| \frac{1}{K} \sum_{k=1}^{K} Regret_{IR}(V^{(k)}; \theta) - \mathbb{E}_{V \sim \mathcal{F}}[Regret_{IR}(V; \theta)] \right| \leq O\left(\sqrt{\frac{\log(1/\delta)}{K}}\right) \quad \textit{with probability } 1 - \delta.$$

This result guarantees that minimizing the empirical regret on a sufficiently large training set allows us to control the true expected regret of the learned mechanism. Combined with the continuity of $p^{\text{OPT-core}}$, these results provide theoretical grounding for our proposed training algorithm. In the next section, we will evaluate the CA-AMA framework and training algorithm empirically.

## 5 Experimental Results

This section presents experimental results that demonstrate the effectiveness of our proposed CA-AMA optimization method across various simulated valuation distributions.

### 5.1 Baselines and Implementation

The main focus is on the comparison between CA-AMA and **Randomized AMA**, represented by LotteryAMA [9] and AMenuNet [14]. We also extend Conditional Auction Net (CAN) [24] to multi-item settings by applying CAN independently to each item, referred to as **Item-CAN**. The classic **VCG** auction [42] and an item-wise application of MyersonNet [16] (denoted **Item-Myerson**) are also included as baselines. GemNet [43] is a menu-based method, which also satisfies strict DSIC

Table 1: Revenue performance of CA-AMA and baseline methods under irregular bidder valuation distributions. CA-AMA consistently outperforms other methods in most scenarios (1.72%, 4.92% average improvements when setting $R_{\text{target}} = 0.001$ and $R_{\text{target}} = 0.01$) and maintains Regret$_{\text{IR}}$ close to the targeted threshold.

| Settings | Item-Myerson | Item-CAN | VCG | Randomized AMA | CA-AMA ($R_{\text{target}} = 0.001$) | | CA-AMA ($R_{\text{target}} = 0.01$) | |
| --- | --- | --- | --- | --- | --- | --- | --- | --- |
| | | | | | Revenue | Regret$_{\text{IR}}$ | Revenue | Regret$_{\text{IR}}$ |
| $2 \times 2$ | 0.5082 | 0.6341 | 0.3911 | 0.6513 | 0.6729 (↑ 3.3%) | 0.0018 | 0.6912 (↑ 6.1%) | 0.0079 |
| $5 \times 2$ | 1.3080 | 1.4376 | 1.3714 | 1.4643 | 1.4938 (↑ 2.0%) | 0.0009 | 1.5525 (↑ 6.0%) | 0.0090 |
| $8 \times 2$ | 1.2077 | 1.6022 | 1.7237 | 1.7645 | 1.8087 (↑ 2.5%) | 0.0009 | 1.8745 (↑ 6.2%) | 0.0083 |
| $10 \times 2$ | 1.4638 | 1.6581 | 1.9106 | 1.9344 | 1.9966 (↑ 3.2%) | 0.0010 | 2.0714 (↑ 7.1%) | 0.0075 |
| $2 \times 3$ | 0.7623 | 0.9511 | 0.5876 | 1.0550 | 1.0512 (↓ 0.4%) | 0.0008 | 1.0870 (↑ 3.0%) | 0.0081 |
| $5 \times 3$ | 1.9619 | 2.1563 | 2.0583 | 2.2682 | 2.2985 (↑ 1.3%) | 0.0017 | 2.3604 (↑ 4.1%) | 0.0085 |
| $8 \times 3$ | 1.8116 | 2.4033 | 2.5844 | 2.6911 | 2.7368 (↑ 1.7%) | 0.0009 | 2.8297 (↑ 5.1%) | 0.0091 |
| $10 \times 3$ | 2.1958 | 2.4871 | 2.8664 | 2.9287 | 2.9893 (↑ 2.1%) | 0.0009 | 3.1033 (↑ 6.0%) | 0.0086 |
| $2 \times 5$ | 1.2705 | 1.5852 | 0.9761 | 1.8508 | 1.8604 (↑ 0.5%) | 0.0011 | 1.8753 (↑ 1.3%) | 0.0078 |
| $5 \times 5$ | 3.2699 | 3.5939 | 3.4286 | 3.7471 | 3.7846 (↑ 1.0%) | 0.0015 | 3.9075 (↑ 4.3%) | 0.0082 |

in theory. We exclude it from our comparisons due to its implementation complexity, especially in multi-item settings.

For implementation, we adopt an over-parameterization strategy for AMA parameters similar to that in AMenuNet [14]. The correlation-aware payment, $p^{\text{Cor}}$, is realized as a three-layer MLP with ReLU activation functions. Identical menu sizes, $|\mathcal{A}|$, are used for Randomized AMA and CA-AMA within the same auction settings to ensure fair comparison. Parameters for the IR regret include target $R_{\text{target}} \in \{0.01, 0.001\}$, initial penalty coefficients $\gamma_0 \in \{3, 6, 8, 10\}$, penalty learning rate $\gamma_\Delta = 0.01$, and a maximum penalty $\bar{\gamma} = 20$. The softmax temperature during mutual training is set to 500. Both mutual training and post-training phases consist of 2,000 iterations, with 32,768 new training samples generated per iteration. A fixed test dataset of 20,000 samples is used for evaluation. Further details on parameter selections are provided in Appendix E.1.

## 5.2 Revenue Performance

We evaluate CA-AMA and baseline methods across several bidder-correlated valuation distributions.

**Irregular Multivariate Normal Distribution.** We adapt the irregular bidder distribution from Huo et al. [24] to a multi-item scenario. Specifically, for each item, the vector of bidders' valuations is drawn with probability 0.5 from one of two multivariate normal distributions. These distributions are constructed using randomly sampled matrices $A_1, A_2 \sim U[-0.2, 0.2]^{n \times n}$ and mean vectors $\mu_1, \mu_2 \sim U[0, 1]^n$. All resulting individual valuations are clamped to the range $[0, 10]$. We generate five distinct sets of distribution parameters $(A_1, A_2, \mu_1, \mu_2)$ and evaluate across various auction scales (number of bidders $n$, number of items $m$).

The average training result is reported in Table 1. Notably, CA-AMA achieves the highest revenue performance in all scenarios. With a target Regret$_{\text{IR}}$ of 0.001 and 0.01, CA-AMA surpasses the best-performing baselines by average margins of 1.72%, 4.92%. Furthermore, CA-AMA consistently maintains Regret$_{\text{IR}}$ near the specified target, even with larger numbers of bidders or items. These results underscore our method's effectiveness in leveraging correlation, even when the underlying correlation structure is complex and not explicitly known to the mechanism.

**Linearly Correlated Valuations.** We investigate scenarios with more explicit linear correlations between bidder valuations. The auction has two bidders, for each item $j$, the valuation of the first bidder, $v_{1j}$, is sampled from $U[0, 1]$. We consider three types of correlation: In **Symmetric Negative**, with probability $\alpha$, $v_{2j} = 1 - v_{1j}$; otherwise, $v_{2j}$ is independently drawn from $U[0, 1]$. In **Symmetric Positive**, with probability $\alpha$, $v_{2j} = v_{1j}$; otherwise, $v_{2j}$ is independently drawn from $U[0, 1]$. In **Asymmetric Negative**, with probability $\alpha$, $v_{2j} = (1 - v_{1j})/4$; otherwise, $v_{2j}$ is independently drawn from $U[0, 1/4]$. Here, $\alpha \in [0, 1]$ controls the correlation strength: $\alpha = 1$ signifies perfect linear correlation, while $\alpha = 0$ indicates bidder independence.

Results for varying $\alpha$ are shown in Figure 2. We observe that Item-CAN achieves optimal revenue when correlation is strong ($\alpha = 1$) but underperforms significantly in bidder-independent scenarios.

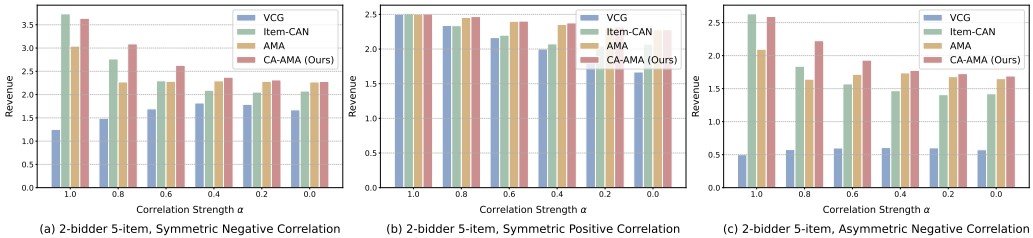

Figure 2: Revenue comparison of CA-AMA against baselines under auctions with explicit linear correlations. Scenarios include: (a) Symmetric Positive Correlation, (b) Symmetric Negative Correlation, and (c) Asymmetric Negative Correlation. $\alpha$ controls the correlation strength: $\alpha = 1$ signifies perfect linear correlation, while $\alpha = 0$ indicates bidder independence.

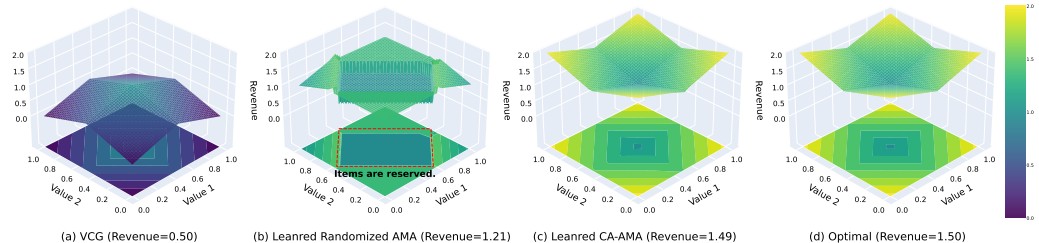

Figure 3: Revenue surfaces of learned CA-AMA and Randomized AMA in a 2-bidder, 2-item perfectly negative linear scenario ($v_{21} = 1 - v_{11}$ and $v_{22} = 1 - v_{12}$). Bidder 1's valuations ($v_{11}, v_{12}$) are on the x-y axes; revenue is on the z-axis. CA-AMA closely approximates the optimal revenue surface, while Randomized AMA often reserves items and has sub-optimal revenue.

Conversely, Randomized AMA performs better in independent scenarios. *In contrast, CA-AMA effectively balances these extremes, automatically leveraging available correlation information.*

Furthermore, *CA-AMA exhibits a more substantial advantage over Randomized AMA in negatively correlated scenarios*, aligning with our theoretical motivation. In positively correlated scenarios with $\alpha = 1$, a simple second-price auction can already extract full surplus. However, in negatively correlated settings, Randomized AMA typically cannot implement payments that decrease with other bidders' valuations (which would be optimal for revenue extraction), thereby limiting its capability.

**Visualization of Randomized AMA and CA-AMA.** In Figure 3, we visualize the revenue surface of the learned CA-AMA and Randomized AMA in a 2-bidder, 2-item **Symmetric Negative** correlation scenario with $\alpha = 1$. The figure plots the extracted revenue (z-axis) as a function of bidder 1's valuations for the two items ($v_{11}$ on the x-axis, $v_{12}$ on the y-axis). CA-AMA's learned revenue surface closely approximates the optimal outcome, demonstrating its ability to learn near-optimal allocation and payment rules. In contrast, Randomized AMA, while an improvement over VCG, deviates significantly from the optimal surface. *Notably, it frequently reserves items even in regions of high valuation, underscoring its inherent limitations in such correlated settings.*

## 6 Conclusion

In this paper, we address the critical limitation of existing AMAs in bidder-correlated settings, where their inherent VCG-style payment rules restrict flexibility and lead to suboptimal revenue extraction. To overcome this, we introduce the CA-AMA, an extended mechanism incorporating an additional correlation-aware payment term. We demonstrate that CA-AMA inherently preserves the DSIC property and can theoretically achieve optimal revenue in single-item auctions under certain correlated distributions where classic AMAs perform arbitrarily poorly. Furthermore, we develop a tailored loss function and a two-stage training algorithm for optimizing CA-AMA, supported by theoretical guarantees on continuity and generalization. Our extensive experimental evaluations across diverse single-item and multi-item auction scenarios confirm the empirical effectiveness of CA-AMA, showcasing its ability to find approximately IR mechanisms and achieve significantly improved revenue compared to AMAs.

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

## Limitation

The main limitation of this work comes from the theoretical results, in which we mainly consider the single-item case. We have discussed the difficulty of analyzing multi-item auctions in the main paper, and we believe this can also serve as a valuable future work.

## A  Detailed Related Work

### A.1  Affine Maximizer Auctions

Affine maximizer auctions (AMAs) generalize the seminal VCG auction by assigning weights to both bidders and allocations, modifying the objective to maximize affine social welfare. Several restricted subclasses of AMA have been studied, including Virtual Valuations Combinatorial Auctions (VVCAs) [30, 31, 39], $\lambda$-auctions [26], mixed bundling auctions [41], and bundling-boosted auctions [2].

The expressiveness of AMAs in comparison to arbitrary auction mechanisms has been formally analyzed in [28]. Beyond expressiveness, algorithmic aspects have also been explored. Sandholm and Likhodedov [39] present optimization methods for finding optimal AMA mechanisms, while Balcan et al. [3, 4] study the sample complexity required to learn such mechanisms. More recently, differentiable optimization techniques have been applied to this setting. For example, LotteryAMA [9] and AMenuNet [14] introduce differentiable approaches to optimize AMA-based auctions using neural networks.

Our work proposes a new framework, CA-AMA, that extends the classical AMA by incorporating bidder correlations. We theoretically characterize its expressiveness relative to traditional AMA in single-item settings and empirically evaluate optimization algorithms for learning revenue-optimal CA-AMA mechanisms across various distributional settings.

### A.2  Differentiable Economics for Auctions

Differentiable economics is a recent and active line of research in automated mechanism design, leveraging neural networks as flexible function approximators and optimizing them using gradient-based methods. Existing work in this area for revenue maximization can be broadly categorized into *characterization-free* and *characterization-based approaches*.

Characterization-free methods do not assume a predefined structure for the mechanism. The foundational work, RegretNet [16], implements the allocation and payment rules as neural networks conditioned on bid profiles. Its loss function jointly optimizes revenue and penalizes violations of DSIC and IR. Building on this, Feng et al. [18] incorporate budget constraints, while Golowich et al. [20] generalize the framework to handle various objectives and constraints. Rahme et al. [37] reframe the design problem as a two-player game with a more efficient loss. Further extensions include PreferenceNet [34], which incorporates fairness preferences, and EquivariantNet [36], a permutation-equivariant architecture tailored for symmetric auctions. Transformer-based methods, such as those

introduced by Ivanov et al. [25] and Duan et al. [13], improve performance in settings with contextual information. Hertrich et al. [22] apply mode connectivity to provide a theoretical explanation for the empirical success of differentiable economics. The combinatorial auction extensions CANet and CAFormer [35] bring these ideas into richer valuation domains.

Characterization-based approaches, by contrast, restrict optimization to a predefined family of mechanisms. AMAs are particularly suitable for this due to their inherent satisfaction of DSIC and IR. LotteryAMA [9] introduces randomized allocation menus over AMA structures, which simplifies optimization. AMenuNet [14] builds upon this with a more expressive architecture and applies it to contextual auctions. Further developments include contextual AMAs for ad auctions [29], dynamic AMA designs [10], and zeroth-order optimization for deterministic AMA mechanisms [15]. In addition, menu-based mechanisms have also been treated with differentiable tools. MenuNet [40] optimizes revenue for single-bidder auctions, while GemNet [43] extends to multi-bidder cases by incorporating over-allocation penalties and post-processing using mixed-integer linear programming.

Our work fits within the characterization-based paradigm. We extend AMA to define CA-AMA, a mechanism that incorporates bidder correlations through a novel correlation-aware payment rule. This new structure retains the theoretical guarantees of classic AMA while significantly improving revenue, both in theory and in practice.

### A.3 Auctions with Bidder Correlations

Modeling bidder correlation is a critical aspect of realistic auction settings. The foundational Crémer-McLean results [7, 8] demonstrate that under certain distributional conditions, it is possible to design mechanisms that are DSIC, interim IR, and extract the full surplus. However, like the Myerson auction, these mechanisms assume full knowledge of the valuation distribution and thus are primarily theoretical.

Subsequent work has relaxed this assumption by exploring scenarios in which the auctioneer has incomplete information. Fu et al. [19], Albert et al. [1], and Yang and Bei [44] study the sample complexity needed to approximate Crémer-McLean-style mechanisms from empirical data. Because computing the optimal mechanism under general correlated settings is NP-hard, approximation algorithms have also been proposed. For instance, Dobzinski et al. [12] design polynomial-time mechanisms that achieve provable approximation guarantees under correlated priors. In contrast, Papadimitriou and Pierrakos [33] and Caragiannis et al. [6] provide upper bounds by constructing distributions where any polynomial-time algorithm performs poorly.

More recent work addresses robustness to correlation. Bei et al. [5] study the correlation-robust design problem, while Zhang [45] and He and Li [21] show that the second-price auction is asymptotically optimal in worst-case correlated environments.

These studies predominantly focus on theoretical designs for single-item auctions. In contrast, our goal is to demonstrate both the theoretical and empirical benefits of CA-AMA in richer combinatorial settings. The most closely related works are Huo et al. [24] and Feldman and Lavi [17]. The former proposes a score-based payment rule, optimized through a max-min neural architecture to approximate optimal revenue in single-item settings. The latter provides a theoretical analysis of the gap between ex-post and ex-interim IR mechanisms, showing that AMA can perform arbitrarily poorly in the presence of correlations. Our results extend this by showing that the performance gap holds even when comparing to ex-post IR mechanisms, and we demonstrate that CA-AMA overcomes this gap.

## B   Omitted Proofs in Section 3

**Proposition 3.1.** *In single-item auctions, for any number of bidders $n$ and any $\epsilon > 0$, there exists a distribution $\mathcal{F}$ such that $REV_{\mathcal{F}}^{D\text{-}AMA} \leq \epsilon \cdot REV_{\mathcal{F}}$. Furthermore, $REV_{\mathcal{F}}^{S\text{-}AMA} < REV_{\mathcal{F}}$ for any menu size $S$.*

*Proof.* See the proof of Theorem 3.3.                                                                                                   □

**Proposition 3.2.** *For any $\mathcal{A}$, $w$, $\lambda$ and correlation-aware function $p^{Cor}$, the CA-AMA mechanism $(g^{CA}, p^{CA})$ satisfies DSIC.*

*Proof.* We verify the DSIC property by definition. For any bidder $i$, its true valuation $\boldsymbol{v}_i$, other bidders' bid $V_{-i}$ and possible bid $\boldsymbol{b}_i$, we define $A_{k^*} := g^{\text{AMA}}(\boldsymbol{v}_i, V_{-i})$, $A_{k'^*} := g^{\text{AMA}}(\boldsymbol{b}_i, V_{-i})$ and $A_{k^*_{-i}} := g^{\text{AMA}}(0, V_{-i})$. Then, we directly compare the utility under truthful report $u_i(\boldsymbol{v}_i, (\boldsymbol{v}_i, V_{-i}))$ and the utility when reporting $\boldsymbol{b}_i$, $u_i(\boldsymbol{v}_i, (\boldsymbol{b}_i, V_{-i}))$. For simplicity, let $V_{-i} = (\boldsymbol{v}_1, \cdots, \boldsymbol{v}_{i-1}, \boldsymbol{v}_{i+1}, \cdots, \boldsymbol{v}_n)$.

$$u_i(\boldsymbol{v}_i, (\boldsymbol{v}_i, V_{-i})) = \boldsymbol{v}_i \cdot (A_{k^*})_i - p_i^{\text{AMA}}(\boldsymbol{v}_i, V_{-i}) - p_i^{\text{Cor}}(V_{-i})$$

$$= \boldsymbol{v}_i \cdot (A_{k^*})_i - \frac{1}{w_i}\left(\sum_{j \neq i} w_j \boldsymbol{v}_j \cdot (A_{k^*_{-i}})_j + \lambda_{k^*_{-i}} - \sum_{j \neq i} w_j \boldsymbol{v}_j \cdot (A_{k^*})_j - \lambda_{k^*}\right) - p_i^{\text{Cor}}(V_{-i})$$

$$= \frac{1}{w_i}\left(\sum_{j=1}^n w_j \boldsymbol{v}_j \cdot (A_{k^*})_j + \lambda_{k^*}\right) - \frac{1}{w_i}\left(\sum_{j \neq i} w_j \boldsymbol{v}_j \cdot (A_{k^*_{-i}})_j + \lambda_{k^*_{-i}}\right) - p_i^{\text{Cor}}(V_{-i})$$

$$\overset{(a)}{\geq} \frac{1}{w_i}\left(\sum_{j=1}^n w_j \boldsymbol{v}_j \cdot (A_{k'^*})_j + \lambda_{k'^*}\right) - \frac{1}{w_i}\left(\sum_{j \neq i} w_j \boldsymbol{v}_j \cdot (A_{k^*_{-i}})_j + \lambda_{k^*_{-i}}\right) - p_i^{\text{Cor}}(V_{-i})$$

$$= \boldsymbol{v}_i \cdot (A_{k'^*})_i - \frac{1}{w_i}\left(\sum_{j \neq i} w_j \boldsymbol{v}_j \cdot (A_{k^*_{-i}})_j + \lambda_{k^*_{-i}} - \sum_{j \neq i} w_j \boldsymbol{v}_j \cdot (A_{k'^*})_j - \lambda_{k'^*}\right) - p_i^{\text{Cor}}(V_{-i})$$

$$= \boldsymbol{v}_i \cdot (A_{k'^*})_i - p_i^{\text{AMA}}(\boldsymbol{b}_i, V_{-i}) - p_i^{\text{Cor}}(V_{-i})$$

$$= u_i(\boldsymbol{v}_i, (\boldsymbol{b}_i, V_{-i})).$$

The inequality (a) is from the definition of $k^*$, $k^* = \arg\max_{k \in [S]} \text{asw}(k; (\boldsymbol{v}_i, V_{-i}))$. $\qquad\square$

**Theorem B.1** (The first part of Theorem 3.3). *In single-item auctions, for any number of bidders $n$: If $\mathcal{F}$ is bidder-independent, then $\text{REV}_{\mathcal{F}}^{\text{D-CA}} = \text{REV}_{\mathcal{F}}^{\text{D-AMA}}$.*

*Proof.* As bidders are independent, we assume that each valuation $\inf\{v_i : v_i \in \text{supp}(\mathcal{F}_i)\} = l_i$ for each $i \in [n]$. *We show that when fixing $\mathcal{A}$ to be set of all deterministic allocations, for an optimal solution $(\boldsymbol{w}, \boldsymbol{\lambda}, (p_i^{\text{Cor}})_{i=1}^n)$ of Problem CA-AMA-OPT, we can construct a feasible solution for Problem AMA-OPT which brings at least the same revenue.* This is sufficient to say that the $\text{REV}^{\text{D-AMA}} \geq \text{REV}^{\text{D-CA}}$.

Let $\mathcal{A}$ be $\{A_0, A_1, A_2, \cdots, A_n\}$, where $A_i$ is the outcome that allocates the item to bidder $i$ and $A_0$ is the outcome that reserves the item. The optimal solution of the CA-AMA is given by $(\boldsymbol{w}, \boldsymbol{\lambda}, (p_i^{\text{Cor}})_{i=1}^n)$. Consider two cases:

If for any $i$ and $\boldsymbol{v}_{-i}$, there is $p_i^{\text{Cor}}(\boldsymbol{v}_{-i}) = 0$, then the revenue of the CA-AMA is equal to the revenue from the AMA parameterized by $(\boldsymbol{w}, \boldsymbol{\lambda})$. *Therefore, below we consider the case that there is at least one $i^*$ and $\boldsymbol{v}_{-i^*}$, such that $p_{i^*}^{\text{Cor}}(\boldsymbol{v}_{-i^*}) > 0$.*

Firstly, the condition $p_{i^*}^{\text{Cor}}(\boldsymbol{v}_{-i^*}) > 0$ means that $g(l_{i^*}, \boldsymbol{v}_{-i^*}; \boldsymbol{w}, \boldsymbol{\lambda}) = A_{i^*}$. Otherwise, the utility of bidder $i^*$ when it realizes its least valuation $l_{i^*}$ is negative, violating the IR constraint. From $g(l_{i^*}, \boldsymbol{v}_{-i^*}; \boldsymbol{w}, \boldsymbol{\lambda}) = A_{i^*}$, we can get the following condition:

$$w_{i^*} l_{i^*} + \lambda_{i^*} > \max\{\max_{j \neq i^*} w_j v_j + \lambda_j, \lambda_0\} \geq \max\{\max_{j \neq i^*} w_j l_j + \lambda_j, \lambda_0\} \geq \lambda_0.$$

Note that this also implies that for any $j \neq i^*$, $p_j^{\text{Cor}}(\boldsymbol{v}_{-j}) \equiv 0$ for any $\boldsymbol{v}_{-j}$. Otherwise, we have $w_{i^*} l_{i^*} + \lambda_{i^*} > w_j l_j + \lambda_j$ and $w_{i^*} l_{i^*} + \lambda_{i^*} < w_j l_j + \lambda_j$ simultaneously.

Secondly, we construct a new AMA based on $(\boldsymbol{w}, \boldsymbol{\lambda})$. Without loss of generality, we set $\lambda_0 = 0$ and define $b := w_{i^*} l_{i^*} + \lambda_{i^*} - \lambda_0 > 0$. The new parameters $(\boldsymbol{w}', \boldsymbol{\lambda}')$ is conducted as $\boldsymbol{w}' = \boldsymbol{w}$, $\lambda_i' = \lambda_i - b$ for all $i \in \{1, 2, \cdots, n\}$, and $\lambda_0' = \lambda_0$.

We analyze the revenue brought by AMA with parameters $(\boldsymbol{w}', \boldsymbol{\lambda}')$. Our goal is to show that for any $\boldsymbol{v} \in \text{supp}(\mathcal{F})$, the payment of the AMA parameterized by $(\boldsymbol{w}', \boldsymbol{\lambda}')$ is at least the payment of the CA-AMA parameterized by $(\boldsymbol{w}, \boldsymbol{\lambda}, (p_i^{\text{Cor}})_{i=1}^n)$.

For any $\boldsymbol{v}$, we obverse that

$$\max_j w_j' v_j + \lambda_j' \geq w_{i^*} v_{i^*} + \lambda_{i^*}' \geq w_{i^*} l_{i^*} + \lambda_{i^*}' = w_{i^*} l_{i^*} + \lambda_{i^*} - b = \lambda_0.$$

574 *Therefore, the item will always be allocated in the new AMA. Furthermore, as the boost variable $\lambda$*
575 *other than $A_0$ changes to the same value, the allocation remains the same.* For this $\boldsymbol{v}$, we consider
576 two cases.

577 1. The item is allocated to bidder $j \neq i^*$.

578 As $p_j^{\text{Cor}} = 0$, the original revenue comes solely from $p_j^{\text{AMA}}$. In new AMA mechanism, the $p_j^{\text{AMA}}$ is
579 computed by:

$$
\begin{aligned}
w_j'\, p_j^{\text{AMA}}(\boldsymbol{v}; \boldsymbol{w}', \boldsymbol{\lambda}') &= \max\{A_0, \max_{k \neq j} w_k' v_k + \lambda_k'\} - \lambda_j' \\
&= \max\{A_0, \max_{k \neq j} w_k v_k + \lambda_k - b\} - \lambda_j + b \\
&\geq \max\{A_0, \max_{k \neq j} w_k v_k + \lambda_k\} - b - \lambda_j + b \\
&= \max\{A_0, \max_{k \neq j} w_k v_k + \lambda_k\} - \lambda_j \\
&= w_j\, p_j^{\text{AMA}}(\boldsymbol{v}; \boldsymbol{w}, \boldsymbol{\lambda}) = w_j'\, p_j^{\text{AMA}}(\boldsymbol{v}; \boldsymbol{w}, \boldsymbol{\lambda}).
\end{aligned}
$$

580 2. The item is allocated to bidder $i^*$.

581 We compare the revenue between $p_{i^*}^{\text{AMA}}(\boldsymbol{v}; \boldsymbol{w}', \boldsymbol{\lambda}')$ and $p_{i^*}^{\text{AMA}}(\boldsymbol{v}; \boldsymbol{w}, \boldsymbol{\lambda}) + p_i^{\text{Cor}}(\boldsymbol{v}_{-i^*})$. Firstly,

$$
\begin{aligned}
w_{i^*}'\, p_{i^*}^{\text{AMA}}(\boldsymbol{v}; \boldsymbol{w}', \boldsymbol{\lambda}') &= \max\{A_0, \max_{k \neq i^*} w_k' v_k + \lambda_k'\} - \lambda_{i^*}' \\
&= \max\{A_0, \max_{k \neq i^*} w_k v_k + \lambda_k - b\} - \lambda_{i^*} + b \\
&\geq \lambda_0 - \lambda_{i^*} + b \\
&= w_{i^*} l_{i^*} + \lambda_{i^*} - \lambda_0 + \lambda_0 - \lambda_{i^*} \\
&= w_{i^*} l_{i^*}.
\end{aligned}
$$

582 For $p_{i^*}^{\text{Cor}}(\boldsymbol{v}_{-i^*})$, by IR constraint, we have,

$$
\begin{aligned}
p_{i^*}^{\text{Cor}}(\boldsymbol{v}_{-i^*}) &\leq l_{i^*} - p_{i^*}^{\text{AMA}}(l_{i^*}, \boldsymbol{v}_{-i^*}; \boldsymbol{w}, \boldsymbol{\lambda}) \\
&= l_{i^*} - \max\{A_0, \max_{k \neq i^*} w_k v_k + \lambda_k\} + \lambda_{i^*} \\
&= l_{i^*} - p_{i^*}^{\text{AMA}}(\boldsymbol{v}; \boldsymbol{w}, \boldsymbol{\lambda}).
\end{aligned}
$$

583 Therefore, $p_{i^*}^{\text{Cor}}(\boldsymbol{v}_{-i^*}) + p_{i^*}^{\text{AMA}}(\boldsymbol{v}; \boldsymbol{w}, \boldsymbol{\lambda}) \leq l_{i^*} \leq p_{i^*}^{\text{AMA}}(\boldsymbol{v}; \boldsymbol{w}', \boldsymbol{\lambda}')$.

584 Hence, for any valuation profile $\boldsymbol{w}$, the revenue by AMA $(\boldsymbol{w}', \boldsymbol{\lambda}')$ is at least the revenue given by
585 CA-AMA $(\boldsymbol{w}, \boldsymbol{\lambda}, (p_i^{\text{Cor}})_{i=1}^n)$. $\qquad\square$

586 **Theorem B.2** (The second part of Theorem 3.3). *In single-item auctions, for any number of bidders $n$*
587 *and any $\epsilon > 0$, there exists a distribution $\mathcal{F}$ such that $\text{REV}_{\mathcal{F}}^{\text{D-AMA}} \leq \epsilon \cdot \text{REV}_{\mathcal{F}}$, while $\text{REV}_{\mathcal{F}}^{\text{D-CA}} = \text{REV}_{\mathcal{F}}$.*
588 *Furthermore, $\text{REV}_{\mathcal{F}}^{\text{S-AMA}} < \text{REV}_{\mathcal{F}}$ for any $S$.*

589 *Proof.* The valuation distribution for the single-item auction is set as follows: Bidder 1's valuation
590 follows a equal revenue distribution on $[\epsilon, 1]$, i.e., the pdf is given by $f(v) = \frac{\epsilon}{(1-\epsilon)v^2}$. The other
591 bidders' valuations are the same and are linear to $v_1$, $v_i = \epsilon_1 \cdot (1 - v_1)$, for all $i \geq 2$. We require
592 $0 < \epsilon_1 < \epsilon < 1$, with specific values to be determined later.

593 Part 1: Showing $\text{REV}_{\mathcal{F}} = \text{REV}_{\mathcal{F}}^{\text{D-CA}}$.

594 For this distribution, it is possible to extract the full social surplus $\max_{i \in [n]} v_i$ as payment for every
595 valuation profile $\boldsymbol{v}$. In the CA-AMA framework, we achieve this by setting: $p_1^{\text{Cor}}(\boldsymbol{v}_{-1}) = (1 - v_2/\epsilon)$,
596 $\mathcal{A}$ to be set of all deterministic allocation, $\boldsymbol{w} = 1$, $\boldsymbol{\lambda}_k = 0$ for all $k \in [S]$. By this, the revenue is the
597 same as first-price auction:

$$
\text{REV}_{\mathcal{F}} = \text{REV}_{\mathcal{F}}^{\text{CA}} = \int_{\epsilon}^{1} f(v) v \, \mathrm{d}v = \int_{\epsilon}^{1} \frac{\epsilon}{(1-\epsilon)v} \, \mathrm{d}v = \frac{\epsilon \ln(1/\epsilon)}{1 - \epsilon}.
$$

598 Part 2: Showing the relationship between $\text{REV}_{\mathcal{F}}^{\text{D-AMA}}$ and $\text{REV}_{\mathcal{F}}$.

In deterministic AMA, $\mathcal{A}$ is fixed to be $\{A_0, A_1, A_2, \cdots, A_n\}$, where $A_i$ is the outcome that allocates the item to bidder $i$ and $A_0$ is the outcome that reserves the item. We first show the following lemma:

**Lemma B.3.** *Under the constructed valuation, for any bidder 1's valuation $v < v'$ and AMA parameter $(\boldsymbol{w}, \boldsymbol{\lambda})$, if bidder 1 wins the item on $v$, then it also wins the item on $v'$.*

*Proof.* When bidder 1's valuation is $v$ and wins the item, we have:

$$w_1 v + \lambda_1 \geq \max\{\lambda_0, \max_{j \geq 2} w_j v_j + \lambda_j\} = \max\{\lambda_0, \max_{j \geq 2} w_j \epsilon_1 (1 - v) + \lambda_j\}.$$

Then for any valuation $v' > v$, we still have that:

$$
\begin{aligned}
w_1 v' + \lambda_1 &> w_1 v + \lambda_1 \\
&\geq \max\{\lambda_0, \max_{j \geq 2} w_j \epsilon_1 (1 - v) + \lambda_j\} \\
&\geq \max\{\lambda_0, \max_{j \geq 2} w_j \epsilon_1 (1 - v') + \lambda_j\}.
\end{aligned}
$$

This means that bidder 1 will also win the item. $\square$

We consider two cases: (1) Bidder 1 never wins the item: then the payment will always be lower than the valuation of other bidders and hence is at most $\epsilon_1$. (2) Bidder 1 does not win when its valuation is less than $v^*$ and wins when its valuation is in $[v^*, 1]$. Still, the payment collected when its valuation is less than $v^*$ is at most $\epsilon_1 \int_\epsilon^{v^*} f(v) dv \leq \epsilon_1$. The payment for the bidder 1 when it wins is bounded by

$$
\begin{aligned}
p_1^{\text{AMA}}(\boldsymbol{v}; \boldsymbol{w}, \boldsymbol{\lambda}) &= \frac{1}{w_1} \left( \max\{\lambda_0, \lambda_1, \max_{j \geq 2} w_j \epsilon_1 (1 - v) + \lambda_j\} - \lambda_1 \right) \\
&\leq \frac{1}{w_1} \left( \max\{\lambda_0, \lambda_1, \max_{j \geq 2} w_j \epsilon_1 (1 - v^*) + \lambda_j\} - \lambda_1 \right) \\
&= p_1^{\text{AMA}}((v^*, \epsilon_1 (1 - \boldsymbol{v}^*)); \boldsymbol{w}, \boldsymbol{\lambda}) \leq v^*.
\end{aligned}
$$

The last inequality is derived from the IR property of any AMA. Therefore, the upper bound of the payment for $[v^*, 1]$ can be computed by:

$$\int_{v^*}^1 v^* f(v) dv = \int_{v^*}^1 v^* \frac{\epsilon}{(1 - \epsilon)v^2} dv = v^* \frac{\epsilon}{(1 - \epsilon)} \left( \frac{1}{v^*} - 1 \right) \leq \frac{\epsilon}{(1 - \epsilon)}.$$

Therefore, the expected payment is bounded by $\frac{\epsilon}{(1-\epsilon)} + \epsilon_1$. As $\text{REV}_{\mathcal{F}} = \frac{\epsilon \ln(1/\epsilon)}{1 - \epsilon}$. For any $\delta$, we can easily set $\epsilon$ and $\epsilon_1$ so that $\text{REV}_{\mathcal{F}}^{\text{D-AMA}} < \delta \cdot \text{REV}_{\mathcal{F}}$.

Part 3: Showing the relationship between $\text{REV}_{\mathcal{F}}^{\text{S-AMA}}$ and $\text{REV}_{\mathcal{F}}$.

As we consider the case that the size of the allocation menu is finite, i.e., $|\mathcal{A}| = S$, $S$ is a constant. Denote the winning allocation as a function to $v_1$, $k(v_1) = \arg\max_{k \in [S]} w_1 v_1 (A_k)_1 + \sum_{j \geq 2} w_j v_j (A_k)_j + \lambda_k = \arg\max_{k \in [S]} w_1 v_1 (A_k)_1 + \sum_{j \geq 2} w_j \epsilon_1 (1 - v_1)(A_k)_j + \lambda_k$. The function must be a piece-wise constant function, and the function has at most $S$ change points by the following lemma.

**Lemma B.4.** *For any $(\mathcal{A}, \boldsymbol{w}, \boldsymbol{\lambda})$, there is at most $S = |\mathcal{A}|$ change points of $k(v_1)$.*

*Proof.* We prove this result by contradiction. Assume that there are $S + 1$ change points, then, there must be a case that for $v_1^1 < v_1^2 < v_1^3$ such that $k = g(v_1^1) = g(v_1^3)$, $k' = g(v_1^2)$, and $k \neq k'$. Then, by definition of AMA's allocation rule, we have

$$w_1 v_1^1 (A_k)_1 + \sum_{j \geq 2} w_j v_j^1 (A_k)_j + \lambda_k \geq w_1 v_1^1 (A_{k'})_1 + \sum_{j \geq 2} w_j v_j^1 (A_{k'})_j + \lambda_{k'} \tag{1}$$

$$w_1 v_1^2 (A_{k'})_1 + \sum_{j \geq 2} w_j v_j^2 (A_{k'})_j + \lambda_{k'} \geq w_1 v_1^2 (A_k)_1 + \sum_{j \geq 2} w_j v_j^2 (A_k)_j + \lambda_k \tag{2}$$

$$w_1 v_1^3 (A_k)_1 + \sum_{j \geq 2} w_j v_j^3 (A_k)_j + \lambda_k \geq w_1 v_1^3 (A_{k'})_1 + \sum_{j \geq 2} w_j v_j^3 (A_{k'})_j + \lambda_{k'}. \tag{3}$$

624 Inserting $v_j = \epsilon_1(1 - v_1)$ $\forall j \geq 2$, by (2) - (1), we have $w_1((A_{k'})_1 - (A_k)_1) \geq$
625 $\epsilon_1 \sum_{j \geq 2} w_j((A_{k'})_j - (A_k)_j)$. By (3) - (2), we have $w_1((A_k)_1 - (A_{k'})_1) \geq \epsilon_1 \sum_{j \geq 2} w_j((A_k)_j -$
626 $(A_{k'})_j)$. The only feasible solution is that $(A_k)_j = (A_{k'})_j$ for all $j \in [n]$, which means $A_k = A_{k'}$
627 and hence brings a contradiction. $\qquad\square$

628 Therefore, we know that there are at most $S$ change points of $g(v_1)$. Suppose these $S' \leq S$ change
629 points are $v_1^0 = \epsilon < v_1^1 < v_1^2 < \cdots < v_1^{S'} < v_1^{S'+1} = 1$ and the corresponding allocations are
630 $A_0, A_1, A_2, \cdots, A_{S'}$. We only consider the interval $[v_1^0, v_1^1)$. If in this interval, $(A_0)_1 < 1$, which
631 means the item is not allocated to bidder 1 deterministically, then the payment loss compared to
632 optimal revenue is at least $(1 - (A_0)_1) \int_{v_1^0=\epsilon}^{v_1^1} (v - \epsilon_1)\mathrm{d}v > 0$.

633 On the other hand, if the allocation satisfies that $(A_0)_1 = 1$. From a similar proof above, we know that
634 the payment in this interval is at most $v_1^0$, which will also results in a gap of $\int_{v_1^0=\epsilon}^{v_1^1} (v - v_1^0)f(v)\mathrm{d}v > 0$
635 compared to the optimal revenue. Therefore, in both cases, we can induce that $\mathrm{REV}_{\mathcal{F}}^{\text{S-AMA}} <$
636 $\mathrm{REV}_{\mathcal{F}}$. $\qquad\square$

# C  Omitted Proofs in Section 4

638 **Theorem 4.1.** *The target function $p_i^{\text{OPT-core}}$ is continuous with respect to the AMA parameters $\mathcal{A}$,*
639 *$\boldsymbol{w}$, and $\boldsymbol{\lambda}$. Furthermore, assume that there exists a constant $C_H > 0$ such that for all $V_{-i}, V'_{-i}$,*
640 *the Hausdorff distance $h(\text{supp}(\mathcal{F}_i(V_{-i})), \text{supp}(\mathcal{F}_i(V'_{-i}))) \leq C_H\|V_{-i} - V'_{-i}\|$, then $p_i^{\text{OPT-core}}$ is also*
641 *continuous with respect to $V_{-i}$.*

642 *Proof.* For simplicity, we use $\phi$ to represent AMA parameters $(\mathcal{A}, \boldsymbol{w}, \boldsymbol{\lambda})$. Specifically, $\mathcal{A} =$
643 $\{A_1, A_2, \cdots, A_S\}$, $\boldsymbol{w} = \{w_1, w_2, \cdots, w_n\}$, and $\boldsymbol{\lambda} = \{\lambda_1, \lambda_2, \cdots, \lambda_S\}$. **For any matrices $A$,**
644 **$A'$ (vectors $v$, $v'$), we denote notation $d_1(A, A')$ $(d_1(v, v'))$ the $L_1$ distance.** For two $\phi$ and $\phi'$,
645 denote
$$d_1(\phi, \phi') = \sum_{k=1}^{S} d_1(A_k, A'_k) + d_1(\boldsymbol{w}, \boldsymbol{w}') + d_1(\boldsymbol{\lambda}, \boldsymbol{\lambda}').$$

646 Recall that $\text{asw}(k; V, \phi)$ is the affine social welfare given by the $k$-th allocation in $\mathcal{A}$, which means:
$$\text{asw}(k; V, \phi) = \sum_{j=1}^{n} w_j(v_j \cdot (A_k)_j) + \lambda_k.$$

647 We first show that $\text{asw}(k; V, \phi)$ is continuous w.r.t $\phi$. For any $\phi$, $\epsilon$, $\phi'$ such that $d_1(\phi, \phi') \leq \epsilon$, and
648 $k \in [S]$, let $\bar{w} := \max_j w_j$, we have

$$
\begin{aligned}
|\text{asw}(k; V, \phi) - \text{asw}(k; V, \phi')| &= |\sum_{j=1}^{n} w_j(v_j \cdot (A_k)_j) + \lambda_k - \sum_{j=1}^{n} w'_j(v_j \cdot (A'_k)_j) - \lambda'_k| \\
&= |\sum_{j=1}^{n} w_j(v_j \cdot (A_k)_j) - \sum_{j=1}^{n} w_j(v_j \cdot (A'_k)_j) \\
&\quad + \sum_{j=1}^{n} w_j(v_j \cdot (A'_k)_j) - \sum_{j=1}^{n} w'_j(v_j \cdot (A'_k)_j) + \lambda_k - \lambda'_k| \\
&\leq \sum_{j=1}^{n} w_j v_j \cdot ((A_k)_j - (A'_k)_j) + \sum_{j=1}^{n} |w_j - w'_j|(v_j \cdot (A'_k)_j) + |\lambda_k - \lambda'_k| \\
&\leq \bar{w} \sum_{j=1}^{n} d_1(A_k, A'_k) + m d_1(\boldsymbol{w}, \boldsymbol{w}') + d_1(\boldsymbol{\lambda}, \boldsymbol{\lambda}') \\
&\leq \max\{\bar{w}, m\} d_1(\phi, \phi') \leq \max\{\bar{w}, m\}\epsilon.
\end{aligned}
$$

649 This means that the continuity of $\phi$ holds.

**(1) The continuity with respect to AMA parameters $\phi$.**

We use asw to compute a bidder's utility under AMA. By the allocation rule and payment rule defined by AMA, there is

$$u_i^{\text{AMA}}(\boldsymbol{v}_i, V; \phi) = \frac{1}{w_i}\left(\max_{k\in[S]} \text{asw}(k; V, \phi) - \max_{k\in[S]} \text{asw}(k; (0, V_{-i}), \phi)\right).$$

And for the target function,

$$p_i^{\text{OPT}-\text{Cor}}(V_{-i}; \phi) = \inf_{\boldsymbol{v}_i \in \text{supp}(\mathcal{F}_i(V_{-i}))} u_i^{\text{AMA}}(\boldsymbol{v}_i, (\boldsymbol{v}_i, V_{-i}); \phi).$$

As both $\max$ and $\inf$ operations do not influence the continuity, we can conclude that $p_i^{\text{OPT}-\text{Cor}}(V_{-i}; \phi)$ is continuous w.r.t. $\phi$ for any $V_{-i}$.

**(2) Continuity in the other bidders' valuations $V_{-i}$.**

Here, the AMA parameters $\phi$ are fixed; we first show that asw is also continuous to $V$. For any $\phi$, $k$, $V$ and $V'$, we have

$$
\begin{aligned}
|\text{asw}(k; V, \phi) - \text{asw}(k; V, \phi)| &= |\sum_{j=1}^{n} w_j(\boldsymbol{v}_j \cdot (A_k)_j) + \lambda_k - \sum_{j=1}^{n} w_j(\boldsymbol{v}'_j \cdot (A_k)_j) - \lambda_k| \\
&= |\sum_{j=1}^{n} w_j(\boldsymbol{v}_j \cdot (A_k)_j) - \sum_{j=1}^{n} w_j(\boldsymbol{v}'_j \cdot (A_k)_j)| \\
&\leq \sum_{j=1}^{n} w_j(|\boldsymbol{v}_j - \boldsymbol{v}'_j| \cdot (A_k)_j) \\
&= \sum_{j=1}^{n} w_j \sum_{t=1}^{m} |\boldsymbol{v}_{jt} - \boldsymbol{v}'_{jt}|(A_k)_{jt} \\
&= \sum_{t=1}^{m} \sum_{j=1}^{n} w_j|\boldsymbol{v}_{jt} - \boldsymbol{v}'_{jt}|(A_k)_{jt} \\
&\leq \sum_{t=1}^{m} \max_j w_j|\boldsymbol{v}_{jt} - \boldsymbol{v}'_{jt}| \\
&\leq \bar{w} d_1(V, V')
\end{aligned}
$$

Then, as the mechanism satisfies DSIC, we will use notation $u_i^{\text{AMA}}(\boldsymbol{v}_i, V_{-i}; \phi)$ to represent the original $u_i^{\text{AMA}}(\boldsymbol{v}_i, (\boldsymbol{v}_i, V_{-i}); \phi)$ as bidders' will always truthfully report. As $u_i^{\text{AMA}}$ is a maximum of a finite number of continuous functions, for any $\boldsymbol{v}_i$, $\boldsymbol{v}'_i$, $V_{-i}$ and $V'_{-i}$,

$$|u_i^{\text{AMA}}(\boldsymbol{v}_i, V_{-i}; \phi) - u_i^{\text{AMA}}(\boldsymbol{v}'_i, V'_{-i}; \phi)| \leq L\, d_1(\boldsymbol{v}_i, \boldsymbol{v}'_i) + L\, d_1(V_{-i}, V'_{-i}), \qquad L := \frac{2\bar{w}}{w_i}. \quad (4)$$

Now, for two valuation profiles $V_{-i}$, $V'_{-i}$, by definition of $p_i^{\text{OPT}-\text{Cor}}$, for any $\epsilon > 0$, we can find a $\boldsymbol{v}_i \in \text{supp}\mathcal{F}_i(V_{-i})$ such that $p_i^{\text{OPT}-\text{Cor}}(V_{-i}) \leq u_i^{\text{AMA}}(\boldsymbol{v}_i, V_{-i}; \phi) \leq p_i^{\text{OPT}-\text{Cor}}(V_{-i}) + \epsilon$. By the Hausdorff assumption on $\text{supp}\mathcal{F}_i(V_{-i})$ and $\text{supp}\mathcal{F}_i(V'_{-i})$, we can find another $\boldsymbol{v}'_i \in \text{supp}\mathcal{F}_i(V'_{-i})$, such that

$$d_1(\boldsymbol{v}_i, \boldsymbol{v}'_i) \leq C_H d_1(V_{-i}, V'_{-i}).$$

Therefore, we can bound the gap in the values

$$
\begin{aligned}
p_i^{\text{OPT}-\text{Cor}}(V_{-i}; \phi) &\geq u_i^{\text{AMA}}(\boldsymbol{v}_i, (\boldsymbol{v}_i, V_{-i}); \phi) - \epsilon \\
&\geq u_i^{\text{AMA}}(\boldsymbol{v}'_i, (\boldsymbol{v}'_i, V'_{-i}); \phi) - L\, d_1(\boldsymbol{v}_i, \boldsymbol{v}'_i) - L\, d_1(V_{-i}, V'_{-i}) - \epsilon \\
&\geq u_i^{\text{AMA}}(\boldsymbol{v}'_i, (\boldsymbol{v}'_i, V'_{-i}); \phi) - L(C_H + 1)\, d_1(V_{-i}, V'_{-i}) - \epsilon \\
&\geq p_i^{\text{OPT}-\text{Cor}}(V'_{-i}; \phi) - \epsilon - L(C_H + 1)\, d_1(V_{-i}, V'_{-i}).
\end{aligned}
$$

667 It is obvious that the vice is also correct, so we can conclude that:

$$|p_i^{\text{OPT}-\text{Cor}}(V_{-i};\phi) - p_i^{\text{OPT}-\text{Cor}}(V'_{-i};\phi)| \leq \epsilon + L(C_H + 1)\,d_1(V_{-i}, V'_{-i})$$
$$= \epsilon + \frac{2\bar{w}}{w_i}(C_H + 1)\,d_1(V_{-i}, V'_{-i}).$$

668 As $\epsilon$ can be chosen sufficiently small, this means that $p_i^{\text{OPT}-\text{Cor}}(\cdot;\phi)$ is $\frac{2\bar{w}}{w_i}(C_H + 1)$-continuous w.r.t.
669 $V_{-i}$ under $L_1$ distance for any fixed $\phi$ under $C_H$-Hausdorff assumption. $\square$

670 **Theorem C.1** (Uniform generalization bound for a 3-layer payment network)**.** *Let $\mathcal{F}$ be an arbitrary*
671 *distribution over valuation profiles $V \in [0,1]^{n \times m}$. For parameters $\theta = (W_1, W_2, W_3)$ satisfying*
672 $\|W_\ell\|_2 \leq M_\ell$ *for $\ell = 1, 2, 3$, consider*

$$Regret_{IR}(V) \;=\; \sum_{i=1}^{n} \max\{0, p_i^{Cor}(V_{-i};\theta) - u_i^{AMA}(\boldsymbol{v}_i, V)\},$$

673 *where the payment network $p_i^{Cor}(\,\cdot\,;\theta) : \mathbb{R}^{(n-1)m} \to \mathbb{R}$ is the depth-3 ReLU network $p_i(x;\theta) =$*
674 $W_3\,\sigma\big(W_2\,\sigma(W_1 x)\big)$ *with $\sigma(z) = \max\{0, z\}$. Let*

$$B_x \;=\; \sqrt{(n-1)m}, \qquad B_p \;=\; B_x \prod_{\ell=1}^{3} M_\ell.$$

675 *For any i.i.d. sample $D = \{V^{(1)}, \ldots, V^{(K)}\} \sim \mathcal{F}^K$ and any confidence level $\delta \in (0,1)$, with*
676 *probability at least $1 - \delta$ (over the draw of $D$) the following inequality holds simultaneously for*
677 every *choice of parameters $\theta$:*

$$\sup_{\theta}\left|\frac{1}{K}\sum_{k=1}^{K} Regret_{IR}(V^{(k)};\theta) - \mathbb{E}Regret_{IR}(V;\theta)\right| \leq 2n\frac{B_p\sqrt{2\log(2d)}}{\sqrt{K}} + nB_p\sqrt{\frac{\log(2/\delta)}{2K}},$$

678 *where $d = \max\{(n-1)m, h_1, h_2, 1\}$ and $h_1, h_2$ are the widths of the first and second hidden layers.*

679 *Proof.* We use $p_i^{\text{Cor}}(V_{-i};\theta)$ and $\text{Regret}_{\text{IR}}(V;\theta)$ to represent the correlation-aware payment and
680 $\text{Regret}_{\text{IR}}$ for input $V$ when the neural network is parameterized by $\theta$. Recall that,

$$\text{Regret}_{\text{IR}}(V;\theta) = \sum_{i=1}^{n} \max\{0, p_i^{\text{Cor}}(V_{-i};\theta) - u_i^{\text{AMA}}(\boldsymbol{v}_i, V)\}.$$

681 Let

$$B_x \;=\; \sqrt{(n-1)m}, \qquad B_p \;=\; B_x \prod_{\ell=1}^{3} M_\ell.$$

682 Since every valuation component lies in $[0,1]$, $\|V_{-i}\|_2 \leq B_x = \sqrt{(n-1)m}$. For ReLU networks,
683 the operator norm is non-expansive, hence,

$$|p_i^{\text{Cor}}(V_{-i};\theta)| \leq \|W_3\|_2 \|W_2\|_2 \|W_1\|_2 \|V_{-i}\|_2 \leq B_p.$$

684 Together with $0 \leq u_i(\boldsymbol{v}_i, V) \leq m$ we therefore have

$$0 \leq \text{Regret}_{\text{IR}}(V;\theta) \leq nB_p.$$

685 Let $\mathcal{P} = \{\text{Regret}_{\text{IR}}(V;\theta) : \theta \in \Theta\}$. By standard symmetrisation (see, e.g., *Bartlett & Mendelson,*
686 *2002*), for any fixed sample $D$

$$\sup_{\theta}\left|\frac{1}{K}\sum_{k=1}^{K}\text{Regret}_{\text{IR}}(V^{(k)};\theta) - \mathbb{E}_{V\sim\mathcal{F}}[\text{Regret}_{\text{IR}}(V;\theta)]\right| \leq 2\,\widehat{R}_K(\mathcal{P}) + nB_p\sqrt{\frac{\log(2/\delta)}{2K}}$$

687 with probability $\geq 1 - \delta$, where $\widehat{R}_K$ is the empirical Rademacher complexity.

Let $\mathcal{P}_i = \{p_i^{\text{Cor}}(V_{-i};\theta) : \theta \in \Theta\}$ be the function class for a single payment component. For a depth-3 ReLU network with spectral-norm bounds $M_\ell$, we have

$$\widehat{R}_K(\mathcal{P}_i) \leq \frac{B_x\left(\prod_{\ell=1}^3 M_\ell\right)\sqrt{2\log(2d)}}{\sqrt{K}},$$

where $d = \max\{(n-1)m, h_1, h_2, 1\}$ and $h_1, h_2$ are the widths of the first and second hidden layers.

Since $\text{Regret}_{\text{IR}}(V;\theta) = \sum_{i=1}^n \max\{0, p_i^{\text{Cor}}(V_{-i};\theta) - u_i^{\text{AMA}}(\boldsymbol{v}_i, V)\}$ and $\max\{0, \cdot\}$ is 1-Lipschitz, we have:

$$\widehat{R}_K(\mathcal{P}) \leq \sum_{i=1}^n \widehat{R}_K(\{p_i^{\text{Cor}}(V_{-i};\theta)\}) = n \cdot \widehat{R}_K(\mathcal{P}_i).$$

Substituting the bound for $\widehat{R}_K(\mathcal{P}_i)$:

$$\widehat{R}_K(\mathcal{P}) \leq n\frac{B_p\sqrt{2\log(2d)}}{\sqrt{K}}.$$

Substituting the complexity estimate for $\widehat{R}_K(\mathcal{P})$, and we finally get:

$$\sup_\theta \left|\frac{1}{K}\sum_{k=1}^K \text{Regret}_{\text{IR}}(V^{(k)};\theta) - \mathbb{E}_{V\sim\mathcal{F}}[\text{Regret}_{\text{IR}}(V;\theta)]\right| \leq 2n\frac{B_p\sqrt{2\log(2d)}}{\sqrt{K}} + nB_p\sqrt{\frac{\log(2/\delta)}{2K}}.$$

$\square$

*Remark* C.2 (Fixed network). If $\theta$ is treated as *fixed* (e.g. after training), *Hoeffding's inequality* immediately gives the simpler bound

$$\left|\frac{1}{K}\sum_k f_{i,\theta}(V^{(k)}) - \mathbb{E}f_{i,\theta}(V)\right| \leq nB_p\sqrt{\frac{\log(2/\delta)}{2K}},$$

so the capacity term vanishes.

# D   Algorithm of CA-AMA

We present the detailed algorithm description for classic randomized AMA optimization methods, including LotteryAMA [9] and AMenuNet [14] in algorithm 1. The two training phases, mutual training and post training, of our CA-AMA are presented in algorithm 2 and algorithm 3, respectively. For the softmax version of AMA, given a valuation profile $V$, the AMA parameters $(\mathcal{A}, \boldsymbol{w}, \boldsymbol{\lambda})$ and temperature $T$, the approximated allocation is calculated as follows,

$$\hat{g}^{\text{AMA}}(V) = \sum_{A\in\mathcal{A}} \frac{e^{\text{asw}(A;V)\cdot T}}{\sum_{A'\in\mathcal{A}} e^{\text{asw}(A';V)\cdot T}} A,$$

$$\hat{g}_{-i}^{\text{AMA}}(V) = \sum_{A\in\mathcal{A}} \frac{e^{\text{asw}_{-i}(A;V)\cdot T}}{\sum_{A'\in\mathcal{A}} e^{\text{asw}_{-i}(A';V)\cdot T}} A.$$

$\text{asw}(k;V)$ is defined as $\sum_{j=1}^n w_j\boldsymbol{v}_j \cdot (A_k)_j + \lambda_k$ and $\text{asw}_{-i}(k;V)$ is $\sum_{j=1,j\neq i}^n w_j\boldsymbol{v}_j \cdot (A_k)_j + \lambda_k$. Based on that, the payment and utility for bidder $i$ is:

$$\hat{p}_i^{\text{AMA}}(V) = \frac{1}{w_i}\left(\text{asw}_{-i}(\hat{g}_{-i}^{\text{AMA}}(V); V) - \text{asw}_{-i}(\hat{g}^{\text{AMA}}(V); V)\right),$$
$$\hat{u}_i^{\text{AMA}}(V) = \boldsymbol{v}_i \cdot \hat{g}^{\text{AMA}}(V)_i - \hat{p}_i^{\text{AMA}}(V). \tag{5}$$

Note that in this approximated version, all operations are differentiable to the AMA parameters $(\mathcal{A}, \boldsymbol{w}, \boldsymbol{\lambda})$. For other notations and equations, please refer to the previous section 4.

---

**Algorithm 1** Classic Randomized AMA Optimization [9, 14]

---

**Require:** Data generator $\mathcal{G}$, initial parameters $\theta$, total iterations $T$, sample size $|S|$.
1: Initialize neural network $p^\theta$ (AMA parameters).
2: Set initial penalty strength $\gamma$.
3: **for** $t = 1$ to $T$ **do**
4:      Generate dataset $S = \{V^1, V^2, \ldots, V^{|S|}\}$ by $\mathcal{G}$.
5:      Get $\mathcal{A}$, $\boldsymbol{w}$, and $\boldsymbol{\lambda}$ from $p^\theta$.
6:      **for** $i = 1$ to $n$ **do**
7:          Approximate AMA payment $\hat{p}_i^{\text{AMA}}$ and utility $\hat{u}_i^{\text{AMA}}$ using softmax by Equation 5.
8:      **end for**
9:      Compute loss:

$$\mathcal{L}(\theta) = \frac{1}{|S|} \sum_{k=1}^{|S|} \sum_{i=1}^{n} -\hat{p}_i^{\text{AMA}}(V^k).$$

10:      Update $p^\theta$ by gradient descent on $\mathcal{L}$.
11: **end for**
**Ensure:** Optimized AMA parameters $p^\theta$.

---

---

**Algorithm 2** Mutual Training of CA-AMA (Ours)

---

**Require:** Data generator $\mathcal{G}$, initial parameters $\theta$, $\phi$, hyperparameters $\gamma$, $\gamma_\Delta$, $R_{\text{target}}$, upper bound $\bar{\gamma}$, total iterations $T$, sample size $|S|$.
1: Initialize neural networks $p^\theta$ (AMA parameters) and $p^\phi$ (correlation-aware payments).
2: Set initial penalty strength $\gamma$.
3: **for** $t = 1$ to $T$ **do**
4:      Generate dataset $S = \{V^1, V^2, \ldots, V^{|S|}\}$ by $\mathcal{G}$.
5:      Get $\mathcal{A}$, $\boldsymbol{w}$, and $\boldsymbol{\lambda}$ from $p^\theta$.
6:      **for** $i = 1$ to $n$ **do**
7:          Approximate AMA payment $\hat{p}_i^{\text{AMA}}$ and utility $\hat{u}_i^{\text{AMA}}$ using softmax by Equation 5.
8:          Get correlation-aware payment $p_i^{\text{Cor}}$ by $p^\phi$.
9:      **end for**
10:      Compute loss:

$$\mathcal{L}(\theta, \phi) = \frac{1}{|S|} \sum_{k=1}^{|S|} \sum_{i=1}^{n} - \left[ \hat{p}_i^{\text{AMA}}(V^k) + p_i^{\text{Cor}}(V_{-i}^k) \right] + \gamma \max\{0, p_i^{\text{Cor}}(V_{-i}^k) - \hat{u}_i^{\text{AMA}}(V^k)\}.$$

11:      Update $p^\theta$, $p^\phi$ by gradient descent on $\mathcal{L}$.
12:      Estimate regret:

$$\tilde{R}(S) = \frac{1}{|S|} \sum_{k=1}^{|S|} \sum_{i=1}^{n} \max\{0, p_i^{\text{Cor}}(V_{-i}^k) - \hat{u}_i^{\text{AMA}}(V^k)\}.$$

13:      Update penalty $\gamma$:

$$\gamma \leftarrow \text{clip}\left(\gamma + \gamma_\Delta (\log \tilde{R}(S) - \log R_{\text{target}}), 1, \bar{\gamma}\right).$$

14: **end for**
**Ensure:** Partially optimized parameters $p^\theta$, $p^\phi$.

---

**Algorithm 3** Post-Training of CA-AMA (Ours)

---

**Require:** Data generator $\mathcal{G}$, parameters $p^\theta$ from mutual training, parameters $\phi$, hyperparameters $\gamma$, $\gamma_\Delta$, $R_{\text{target}}$, upper bound $\bar{\gamma}$, total iterations $T$, sample size $|S|$.
 1: Freeze neural network $p^\theta$.
 2: **for** $t = 1$ to $T$ **do**
 3:     Generate dataset $S = \{V^1, V^2, \dots, V^{|S|}\}$ by $\mathcal{G}$.
 4:     Get $\mathcal{A}$, $\boldsymbol{w}$, and $\boldsymbol{\lambda}$ from $p^\theta$.
 5:     **for** $i = 1$ to $n$ **do**
 6:         Compute exact AMA payment $p_i^{\text{AMA}}$ and utility $u_i^{\text{AMA}}$ using true $\arg\max$.
 7:         Get correlation-aware payment $p_i^{\text{Cor}}$ by $p^\phi$.
 8:     **end for**
 9:     Compute loss:

$$\mathcal{L}(\phi) = \frac{1}{|S|} \sum_{k=1}^{|S|} \sum_{i=1}^{n} - \left[ p_i^{\text{AMA}}(V^k) + p_i^{\text{Cor}}(V_{-i}^k) \right] + \gamma \max\{0, p_i^{\text{Cor}}(V_{-i}^k) - u_i^{\text{AMA}}(V^k)\}.$$

10:     Update $p^\phi$ by gradient descent on $\mathcal{L}$.
11:     Estimate regret $\tilde{R}(S)$.
12:     Update penalty $\gamma$:

$$\gamma \leftarrow \text{clip}\left( \gamma + \gamma_\Delta (\log \tilde{R}(S) - \log R_{\text{target}}), 1, \bar{\gamma} \right).$$

13: **end for**
**Ensure:** Fully optimized parameters $p^\phi$.

---

Table 2: Hyperparameters and training times of CA-AMA and Randomized AMA methods.

| Hyperparameter | 2×2 | 5×2 | 8×2 | 10×2 | 2×3 |
|---|---|---|---|---|---|
| Initial penalization term $\gamma_0$ | 3 | 6 | 6 | 8 | 5 |
| Menu size $|\mathcal{A}|$ | 32 | 64 | 128 | 256 | 64 |
| CA-AMA training time (min) | 20 | 26 | 40 | 47 | 22 |
| AMenuNet training time (min) | 19 | 23 | 33 | 40 | 20 |

| Hyperparameter | 5×3 | 8×3 | 10×3 | 2×5 | 5×5 |
|---|---|---|---|---|---|
| Initial penalization term $\gamma_0$ | 6 | 8 | 8 | 3 | 10 |
| Menu size $|\mathcal{A}|$ | 1024 | 2048 | 2048 | 256 | 2048 |
| CA-AMA training time (min) | 40 | 80 | 90 | 27 | 70 |
| AMenuNet training time (min) | 40 | 75 | 85 | 24 | 65 |

# E   Further Experimental Descriptions

## E.1   Implementation Details

Most hyperparameters are the same for all settings, as we have introduced in section 5. Only two hyperparameters vary for different settings: the initial penalization term $\gamma_0$ and the menu size $|\mathcal{A}|$. We present the choices taken in our experiments, and also present the total training time for different auction settings ($n$ and $m$). As the implementation of CA-AMA only adds a computation for the $\text{Regret}_{\text{IR}}$ term and the correlation-aware payment is represented by simply a three-layer MLP, the training time does not significantly increase compared to [14].

## E.2   Further Experimental Results

We consider a 2-bidder single-item auction setting. The two bidders are also linearly correlated: the first bidder's valuation $v_1$ is sampled from an equal revenue distribution clamped within $[\epsilon, 1]$. The second bidder's valuation $v_2$ equals to $\frac{\epsilon}{1-\epsilon}(1 - v_1)$. To make the outcome significant, we multiply

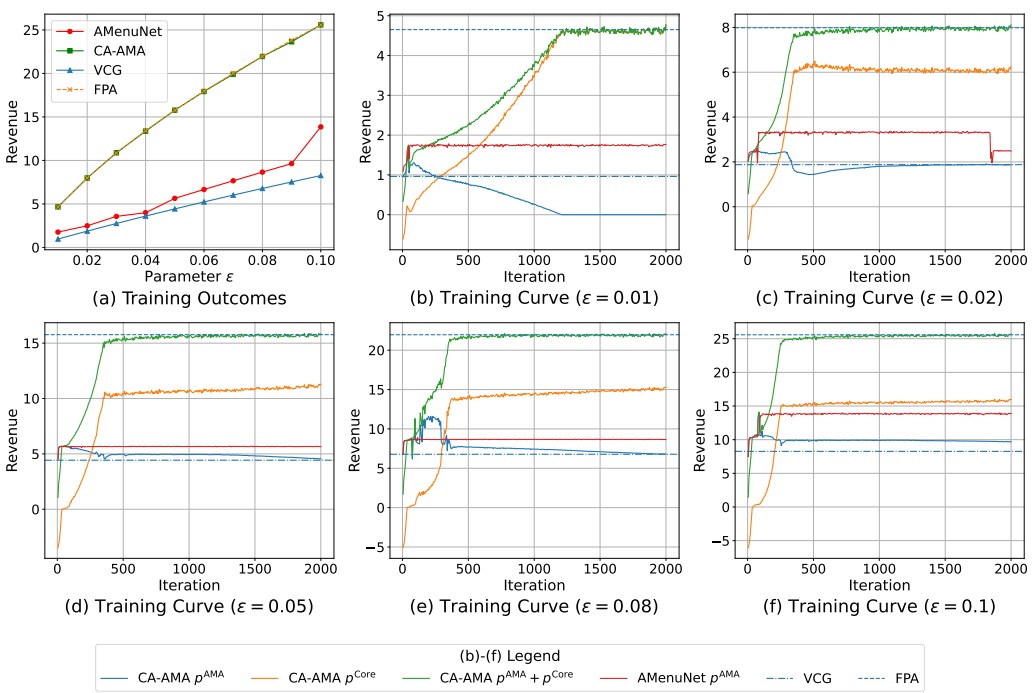

Figure 4: The revenue results and training curves of CA-AMA and Randomized AMA (implemented by AMenuNet [14]) in auctions with the first bidder's valuation $v_1$ following equal revenue distribution on $[\epsilon, 1]$ and the second bidder's valuation $v_2 = \frac{\epsilon}{1-\epsilon}(1 - v_1)$. As the Regret$_{\text{IR}}$ in all cases is less than $1e - 5$, it is not plotted in the figure.

all valuations by $100$. This is the case we constructed in the proof of theorem 3.3, where we prove that when $\epsilon$ is sufficiently small, then the optimal revenue obtained by randomized AMA can be arbitrarily poorer than optimal CA-AMA.

Different values of $\epsilon$ are selected, ranging from $0.01$ to $0.1$. We present the results for different $\epsilon$ and plot the training curves for some cases in Figure 4. As for comparison, the revenue gained by VCG and FPA (First Price Auction), which extracts the full surplus and hence represents the optimal revenue, and the revenue obtained by optimal Randomized AMA, are also plotted. As is demonstrated in the figure, CA-AMA succeeds in reaching the optimal revenue, significantly surpassing Randomized AMA. From the dynamics of payment $p^{\text{Cor}}$ and $p^{\text{AMA}}$, we observe that CA-AMA can effectively tell the correlation information in this distribution and hence $p^{\text{Cor}}$ dominates in all cases. Compared to Randomized AMA, although the revenue part comes from AMA (CA-AMA $p^{\text{AMA}}$) is less than AMenuNet $p^{\text{AMA}}$, the total revenue CA-AMA $p^{\text{AMA}} + p^{\text{Cor}}$ is significantly higher than it.

### E.3 Influence of the Target Regret

This section investigates the impact of the target level of IR regret, $R_{\text{target}}$, on the revenue achieved by our optimized CA-AMA mechanism. Experiments are conducted in a 2-bidder 2-item auction setting with irregular multivariate normal value distributions, as described in detail in Section 5. We evaluate $R_{\text{target}}$ for values in the set $\{0.05, 0.02, 0.01, 0.005, 0.002, 0.001, 0.0005, 0.0001\}$. Figure 5 presents the average revenue and the achieved IR regret over $5$ independent test runs for CA-AMA at each target regret level. For comparison, the revenue achieved by Randomized AMA, VCG, and Item-CAN is also included.

Firstly, we observe that after training, the achieved IR regret for CA-AMA is consistently close to the specified target value, even for very small targets like $R_{\text{target}} = 0.0001$. This demonstrates the effectiveness of our training algorithm in steering the mechanism towards a desired level of IR

compliance, mitigating the significant IR violations that can occur with standard AMA approaches. Secondly, as $R_{\text{target}}$ approaches $0$, the revenue obtained by CA-AMA tends to decrease. Nevertheless, CA-AMA consistently yields higher average revenue than Randomized AMA across all tested target regret levels.

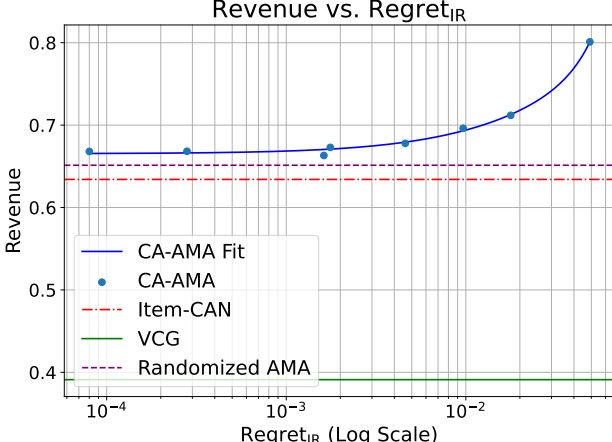

Figure 5: Average revenue vs. achieved IR regret for the optimized CA-AMA under different target IR regret ($R_{\text{target}}$). Results are averaged over $5$ test runs in a 2-bidder, 2-item auction setting with irregular multivariate normal value distributions. Revenue obtained by Randomized AMA, VCG, and Item-CAN is included for comparison.

