# OpenReview forum: "Enhancing Affine Maximizer Auctions with Correlation-Aware Payment"
_NeurIPS.cc/2025/Conference — Submitted to NeurIPS 2025_

### Official Review · Reviewer_TSDr · 2025-06-15

**Clarity:** 3
**Significance:** 3
**Originality:** 3
**Rating:** 4
**Confidence:** 2

**Summary:**

This paper introduces Correlation-Aware AMA (CA-AMA), being a modification of affine-maximizer auctions (AMA) that adds a payment term for each bidder that depends on the valuations of all other bidders. The authors nicely motivated the need for enhancing AMA’s expressiveness and transitioning from bidder-independent settings to bidder-correlated settings. The authors proved (under certain assumptions) that their method satisfies DSIC, but IR can be high.
This motivated the formulation of an optimization framework to maximize revenue subject to IR.  The authors prove in Theorem 3.3 the following under single-item auctions: when bidders are independent, CA-AMA offers no advantage over classic AMA, but there exist correlated distributions  CA-AMA can achieve optimal revenue, whereas AMAs fail. Unfortunately, this analysis formally holds only for single-item auctions.

The authors also presented training methods: a two-stage gradient-based algorithm with a loss that balances revenue against IR-violation.  Theorem 4.1 states that, under certain assumptions, the optimal correlation-payment function $p^{\text{OPT}}_{\text{Cor}}(V^{-i})$ is continuous in the AMA parameters and $V^{-i}$. The authors also present in Theorem 4.2 a Rademacher-complexity bound for IR-regret.  Empirical results on synthetic correlated distributions (Gaussian mixtures withlinear correlations) illustrate that CA-AMA gets noticeably improved revenue than several baseline methods while having IR-regret near target.

**Questions:**

See weaknesses.

**Ethical Concerns:**

["NO or VERY MINOR ethics concerns only"]

**Final Justification:**

Thank you for your detailed reply.

After reading all other reviewers' comments and the authors' replies, I decided to keep my rating and support accepting the paper. Again, I should note that I'm not considering myself an expert in that field.

Specifically,
- The authors offer an approach to transform our trained CA-AMA into a strictly IR mechanism, addressing W1.
- The authors addressed my concerns regarding W2-W4. Thank you.
- I appreciate the discussion on W5. Indeed, it would be valuable to have theoretical results concerning AMA and CA-AMA.

**Limitations:**

The authors discussed the limitations of the work in the appendix. I’d also include the need to know the true valuations for training, and issues with robustness to train/test distribution shifts, and most importantly -- violations of IR.

**Quality:**

3

**Strengths And Weaknesses:**

Below I give a list of strengths and weakeness that I wanted to reflect; I should note that I’m not an expert in this field.

**Strengths**

S1. Novelty: As far as I understand, the idea of including a correlation-dependent payment term is new in the AMA literature.

S2. Theory: The authors provide a good theoretical basis for their proposed method: DSIC (Prop. 3.3), optimal revenue in a special case (Theorem 3.3), continuity (Theorem 4.1), and generalization bound (Theorem 4.2).

S3. Well-motivated optimization method for CA-AMA.

S4. Empirical analysis shows consistent gains in revenue, although somewhat modest.

**Weaknesses**

W1 (main concern). The proposed mechanism is only approximately IR by design. How do violations of IR affect the practical use of the proposed method? It seems unreasonable for a bidder to participate in an auction where they might end up with negative utility: it means a bidder could end up paying more than their actual bid, which I find unjustified.

W2. I wonder how the method performs on real data, and how the limited expressivity of standard AMA is problematic in practice. Currently, the experiments are conducted using simulated data, which is valuable but not necessarily representative of real-world bidder-correlation structures.

W3. Related to the above, are there standard errors for Table 1? Same for Figure 2 and Figure 5.

W4. Still in the context of the experiments, the authors compared their approach to AMA methods and classical mechanisms, but did not compare it to the popular RegretNet method. The authors argue in the discussion in the appendix that RegretNet is not DSIC and thus such a comparison is less appropriate. But, on the other hand, RegretNet is IR in contrast with the proposed CA-AMA method. So, which method should one use?

W5. The theory holds under a single-item assumption, but I understand the challenge of moving beyond this setup.

---

> ### Author Rebuttal · Authors · 2025-07-31
>
> We thank the reviewer for the insightful feedback and constructive suggestions. We address the specific points below.
>
> ---
> **Weakness 1**: About the violation of IR. The proposed mechanism is only approximately IR by design. How do violations of IR affect the practical use of the proposed method? It seems unreasonable for a bidder to participate in an auction where they might end up with negative utility: it means a bidder could end up paying more than their actual bid, which I find unjustified.
>
> **Response**: We acknowledge that the violation of IR, or IR-regret, may influence bidder trust in the system. However, we would like to clarify that **the impact is relatively minimal within the framework we propose.**
>
> Firstly, as shown in our experimental results, our two-phase optimization ensures that IR-regret converges to the target regret value in most cases. This indicates that IR-regret is relatively controllable, and we can adjust its value according to different needs.
>
> Secondly, with low IR-regret, **we offer a straightforward approach to transform our trained CA-AMA into a strictly IR mechanism:** we provide each bidder with the option to opt out (without payment or allocation) after seeing their allocation and payment. This ensures that only bidders with positive utility remain, while those with negative utility will choose to quit. As a result, the modified mechanism satisfies strict IR. This adjustment does not affect the optimality of truthful reporting, maintaining both DSIC and IR properties.
>
> We will include additional experimental results to demonstrate this modification in the final version of the paper.
>
> * 2-bidder 5-item, Symmetric Negative Correlation
>
> | Correlation Strength $\alpha$ | 1      | 0.8    | 0.6    | 0.4    | 0.2    | 0       |
> |-------------------------------|--------|--------|--------|--------|--------|---------|
> | VCG                           | 1.2507 | 1.4899 | 1.6933 | 1.8192 | 1.7901 | 1.6714  |
> | Item-CAN                      | 3.7345 | 2.7655 | 2.2955 | 2.0925 | 2.0535 | 2.076   |
> | AMA                           | 3.0413 | 2.2706 | 2.2857 | 2.2947 | 2.2848 | 2.261   |
> | CA-AMA Revenue                | 3.6378 | 3.0886 | 2.6283 | 2.373  | 2.3145 | 2.282   |
> | CA-AMA Regret                 | 0.0008 | 0.0018 | 0.0017 | 0.0011 | 0.001  | 0.0008  |
> | CA-AMA (Strict IR)            | 3.5584 | 2.9638 | 2.5607 | 2.3534 | 2.3008 | 2.262   |
>
>
> * 2-bidder 5-item, Asymmetric Negative Correlation
>
> | Correlation Strength $\alpha$ | 1      | 0.8    | 0.6    | 0.4    | 0.2    | 0       |
> |-------------------------------|--------|--------|--------|--------|--------|---------|
> | VCG                           | 0.5    | 0.5772 | 0.6008 | 0.6068 | 0.6025 | 0.574   |
> | Item-CAN                      | 2.6325 | 1.839  | 1.5715 | 1.4685 | 1.408  | 1.422   |
> | AMA                           | 2.097  | 1.6431 | 1.7174 | 1.7377 | 1.6806 | 1.651   |
> | CA-AMA Revenue                | 2.5919 | 2.2257 | 1.9313 | 1.7638 | 1.7268 | 1.6911  |
> | CA-AMA Regret                 | 0.0017 | 0.0015 | 0.0019 | 0.0013 | 0.001  | 0.0009  |
> | CA-AMA (Strict IR)            | 2.53   | 2.132  | 1.8734 | 1.7573 | 1.7135 | 1.6821  |
>
> Note that "CA-AMA (Strict IR)" refers to the mechanism after post-modification, which satisfies both DSIC and strict IR. As shown in the tables, CA-AMA continues to outperform other baseline methods in most scenarios.
>
> Finally, as defined in Definition 2.2, **our IR is stronger than Bayesian-IR and closer to ex-post IR.** This means our CA-AMA may not violate Bayesian-IR. In practical scenarios, bidders may only be aware of their own valuation and compute the expected utility based on the uncertainty of others. As a result, the expected utility can still be positive, even when the realized utility is negative. **Because the expected utility is positive, bidders would still participate in the auction.**
>
> ---
> **Weakness 2**: About the experiments on real data.
>
> **Response**: We believe that our method is also applicable to real data distributions. However, real data is often held by industrial companies and is not readily available for research purposes. Therefore, in line with previous studies, we conducted experiments using various synthetic distributions.
>
> ---
> **Weakness 3**: About the standard errors for Table 1, Figure 2, and Figure 5.
>
> **Response**: Overall, the standard error is relatively small (generally less than 5\% of the revenue performance). We will include this information in the final version of the paper to provide clarity.
>
> ---
> **Weakness 4**: About the comparison with RegretNet.
>
> **Response**: Our primary innovation lies in improving the classic AMA within DSIC mechanisms. As such, we did not include RegretNet in our experiments. Moreover, **the paper by [1] on AMenuNet already demonstrated that the classic randomized AMA achieves competitive results compared to RegretNet.** Since AMenuNet is used as the baseline in our paper, this sufficiently demonstrates that our CA-AMA is also competitive with RegretNet.
>
> We will add **further discussion on the comparison between CA-AMA, which introduces IR-regret, and RegretNet, which introduces IC-regret.** In theory, RegretNet has the potential to express the universal optimal auction, while CA-AMA is still limited in its expressiveness. Nonetheless, CA-AMA has several advantages:
>
> 1. IR-regret computation: **Calculating IR-regret is significantly easier than computing IC-regret**. IR-regret only requires the resulting payment and valuation of an auction. In contrast, IC-regret needs additional sampled bids to compute an approximate best bid aside from truthful reporting, which is time-consuming during training due to the need for more auction results.
>
> 2. Post-processing flexibility: **IR-regret is easier to adjust**. As noted in our response to Weakness 1, if a low IR-regret or zero regret is desired, we provide simple methods to achieve this goal. RegretNet, on the other hand, faces difficulties in adjusting the degree of IC violation without further training.
>
> 3. Interpretability: **Since CA-AMA is based on AMA, it is more interpretable**. The allocation menu, bidder weights, and allocation boosts are transparent.
>
> Overall, we believe **CA-AMA strikes a balance between empirical performance, theoretical properties, and implementation efficiency.**
>
> ---
> **Weakness 5**: About the theoretical results.
>
> **Response**: We would like to add more discussion regarding the difficulty of proving our conjecture that "CA-AMA = AMA in bidder-independent auctions" (the first part of Theorem 3.3) for multi-item auctions. We are attempting a proof by contradiction, constructing an AMA that achieves at least the same revenue as CA-AMA.
>
> Firstly, the optimal revenue in multi-item settings is generally unknown and remains a fundamental open problem in game theory. Even for AMA, there are no existing results that characterize the optimal AMA in general valuation distributions. This makes it challenging to utilize the properties of the optimal AMA.
>
> Secondly, in multi-item auctions, the allocation of one item can depend on the allocation of others. For example, an item might be reserved if bidders' valuations for other items are low. This introduces significant differences between the single-item and multi-item cases, resulting in more discontinuities in allocations.
>
> Therefore, we primarily validate the performance of CA-AMA in multi-item settings empirically. We believe that further theoretical results concerning AMA and CA-AMA are valuable areas for future work.

---

> > ### Comment · Reviewer_TSDr · 2025-08-01
> >
> > Thank you for your detailed reply. I've edited my initial review. I decided to keep my positive rating, recommending acceptance.

---

### Official Review · Reviewer_Csj4 · 2025-06-21

**Clarity:** 3
**Significance:** 2
**Originality:** 3
**Rating:** 4
**Confidence:** 4

**Summary:**

There has been many recent works on differentiable economics focusing on the affine maximizer auctions (AMA). The key idea of this literature is to "model an auction as a multi-layer
neural network, frame optimal auction design as a constrained learning problem", and then solve the said problem "using standard machine learning pipeline" (Duetting et al 2017)[Optimal Auctions through Deep Learning: Advances in Differentiable Economics]. This paper considers a variant of the AMA with additional parameters designed to maximize revenue better with interdependent bids. It builds on the ideas of Cr\'{e}mer and McLean (1987), to augment the VCG pivot payment by a term that depends only on other bids, called the correlation-aware payment. This preserves DSIC but not IR (individual rationality), so a regret penalty on IR is added.

The paper's main contributions are the following.

1. A constructed example to demonstrate that VCG-style AMA can fail to optimize revenue when bids are sufficiently negatively correlated.
2. Introduce an additional term in the VCG payment in the style of Cremer-McLean, generalizing it.
3. Add a regret penalty to the standard AMA revenue optimization to encourage the IR constraint, in th style of Duetting et al.
4. Give some theoretical results to show that the new optimization problem (CA-AMA) has some reasonable properties (continuous w.r.t. parameters, a K^{1/2} regret bound).
5. Demonstrate experimentally that CA-AMA stays close to the targetted Regret threshold (ie keeps IR violation within a specified amount). Though CA-AMA is not optimal in close to $\pm 1$ correlation, it is better than AMA in strongly correlated settings and seems less sensitive to correlation strength compared to other methods (eg ItemCAN). AMA is better in independent settings.

**Questions:**

Minor comment: I find it a bit hard to keep track of all of the competitor's methods and assumptions, so could you please mention in your results table which of the competing methods also have a non-zero regret for IR (and if so how much), and which guarantees IR ? This way we can have a fair comparison, ie that the new method CA-AMA is not obtaining better revenue at the expense of loosening the constraints. Probably showing the loss (-revenue + regret) would help, I would expect CA-AMA to be optimal there, but it would be helpful to have an understanding such as "for every X unit of revenue gained, some Y united of regret must be paid for".

**Ethical Concerns:**

["NO or VERY MINOR ethics concerns only"]

**Limitations:**

Yes

**Quality:**

3

**Strengths And Weaknesses:**

Strengths: natural setup (interdependent values), decent solution (Cremer-McLean inspiration), good exposition.

Weaknesses: experimental results show minor improvements over Randomized AMA except for strongly correlated (negative or positive) values, which might not be too realistic. It seems unlikely to me for this to dislodge randomized AMA as the standard benchmark in practice, so from that view the literature impact could be limited.

---

> ### Author Rebuttal · Authors · 2025-07-31
>
> We thank the reviewer for the insightful feedback and constructive suggestions. We address the specific points below.
>
> ---
> **Weakness**: Minor improvements over Randomized AMA except for strongly correlated (negative or positive) values.
>
> **Response:** We appreciate the opportunity to clarify the contributions of CA-AMA, particularly concerning its performance in scenarios with varying degrees of correlation.
>
> First, it is crucial to understand that in distributions with lower correlation, **classic AMA and CA-AMA may indeed exhibit similar expressiveness.** Our theoretical results, specifically **Theorem 3.3, demonstrate that "optimal CA-AMA = optimal AMA" in single-item, bidder-independent cases.** We have also observed that this property holds even in cases with correlation, provided the support of each conditional distribution remains the same. Specifically, for a bidder $i$, if the support satisfies $\mathrm{supp}(\mathcal{F}\_i(V_{-i})) \equiv S$ for all realizations of $V_{-i}$, the result continues to apply. We further conjecture that this result extends to the multi-bidder setting. Therefore, the minor improvement in such scenarios is not a weakness but a theoretically guaranteed outcome, which itself is a valuable part of our contribution.
>
> Second, we respectfully argue that **strong correlations are, in fact, plausible in real-world scenarios.** Take online advertising, for example, where advertisers often have completely opposing preferences. In such contexts, CA-AMA offers a more flexible and powerful tool to leverage this correlation information compared to AMA. As is shown in the Figure 2, and Figure 4, 5 in the Appendix, CA-AMA significant improvement in performance.
>
> Finally, from a practical implementation standpoint, the optimization process for CA-AMA is only **marginally more complex than AMA's.** The increase in training time is minimal (as shown in Table 2), and performance remains stable across all settings. More importantly, CA-AMA consistently outperforms AMA in nearly all scenarios. When compared to methods like RegretNet, which requires computing IC-regret and offers only approximate IC, and GemNet, which uses Integer Programming to ensure allocation feasibility, we believe that **CA-AMA strikes a better balance among empirical performance, theoretical properties, and and implementation efficiency.**
>
> ---
> **Question**: Which of the competing methods also have a non-zero regret for IR (and if so, how much), and which guarantees IR? Probably showing the loss (-revenue + regret) would help, I would expect CA-AMA to be optimal there, but it would be helpful to have an understanding such as "for every X unit of revenue gained, some Y united of regret must be paid for".
>
> **Response**: First, we would like to clarify that **all competing methods, including Item-Myerson, Item-CAN, VCG, and Randomized AMA, guarantee strict IR,** meaning they do not incur IR regret. Our innovation introduces a conceptual shift, where we allow for the loss of IR during training, integrating it into the optimization goal. With different preset IR-regret targets, our implementation ensures that the achieved IR-regret is close to the target, while still maximizing revenue performance.
>
> Additionally, to the best of our knowledge, no previous work has characterized the trade-off between IR-regret and revenue in such detail. **We also offer a straightforward approach to modify our trained CA-AMA into a strictly IR mechanism:** we add an additional option for each bidder, allowing them to opt out (without payment or allocation) after viewing their allocation and payment. This ensures that only bidders with positive utility will stay, while those with negative utility will choose to quit. Consequently, the modified mechanism satisfies strict IR. Since this modification does not affect the optimality of truthful reporting, the mechanism remains DSIC and IR.
>
> We will add further experimental results to the final version of the paper to illustrate this modification:
>
> * 2-bidder 5-item, Symmetric Negative Correlation
>
> | Correlation Strength $\alpha$ | 1      | 0.8    | 0.6    | 0.4    | 0.2    | 0       |
> |-------------------------------|--------|--------|--------|--------|--------|---------|
> | VCG                           | 1.2507 | 1.4899 | 1.6933 | 1.8192 | 1.7901 | 1.6714  |
> | Item-CAN                      | 3.7345 | 2.7655 | 2.2955 | 2.0925 | 2.0535 | 2.076   |
> | AMA                           | 3.0413 | 2.2706 | 2.2857 | 2.2947 | 2.2848 | 2.261   |
> | CA-AMA Revenue                | 3.6378 | 3.0886 | 2.6283 | 2.373  | 2.3145 | 2.282   |
> | CA-AMA Regret                 | 0.0008 | 0.0018 | 0.0017 | 0.0011 | 0.001  | 0.0008  |
> | CA-AMA (Strict IR)            | 3.5584 | 2.9638 | 2.5607 | 2.3534 | 2.3008 | 2.262   |
>
>
> * 2-bidder 5-item, Asymmetric Negative Correlation
>
> | Correlation Strength $\alpha$ | 1      | 0.8    | 0.6    | 0.4    | 0.2    | 0       |
> |-------------------------------|--------|--------|--------|--------|--------|---------|
> | VCG                           | 0.5    | 0.5772 | 0.6008 | 0.6068 | 0.6025 | 0.574   |
> | Item-CAN                      | 2.6325 | 1.839  | 1.5715 | 1.4685 | 1.408  | 1.422   |
> | AMA                           | 2.097  | 1.6431 | 1.7174 | 1.7377 | 1.6806 | 1.651   |
> | CA-AMA Revenue                | 2.5919 | 2.2257 | 1.9313 | 1.7638 | 1.7268 | 1.6911  |
> | CA-AMA Regret                 | 0.0017 | 0.0015 | 0.0019 | 0.0013 | 0.001  | 0.0009  |
> | CA-AMA (Strict IR)            | 2.53   | 2.132  | 1.8734 | 1.7573 | 1.7135 | 1.6821  |
>
> Note that the "CA-AMA (Strict IR)" refers to the mechanism after post-modification, which satisfies both DSIC and strict IR. As shown in the table, CA-AMA outperforms the baseline methods in most scenarios.

---

> ### Author Response · Authors · 2025-08-06
>
> Dear Reviewer Csj4,
>
> Thank you once again for your time and effort in providing valuable comments on our paper. As the author-reviewer discussion phase approaches its conclusion, we would like to kindly confirm whether we have sufficiently addressed all (or at least some) of your concerns. In particular, **we have provided additional explanations regarding the empirical improvement of our method and the balance between IR-regret and revenue performance.**
>
> Should any questions remain or if further clarification is needed, please do not hesitate to let us know. If you are satisfied with our responses, we would greatly appreciate your consideration in adjusting the evaluation scores accordingly.
>
> We sincerely look forward to your feedback.

---

### Official Review · Reviewer_YyPf · 2025-07-03

**Clarity:** 4
**Significance:** 2
**Originality:** 2
**Rating:** 4
**Confidence:** 2

**Summary:**

This paper proposes a novel family of DSIC and IR multi-item auctions with additive valuations that generalizes Affine Maximizer Auctions (AMAs), a widely used auction family in automated mechanism design. The paper focuses on the Bayesian auction setting with additive valuations: there are many items and many buyers, each buyer has a private value for each item, and buyers’ valuations are drawn from a possibly correlated prior distribution. The goal is to design individually rational (IR) and dominant strategy incentive compatible (DSIC) mechanisms that maximize expected revenue. In general, finding the optimal IC mechanism in the multi-item setting is notoriously difficult. A common approach is to restrict to a specific family of IC and IR mechanisms and optimize within that family for the given instance. One notable example is the AMA, an extension of the VCG mechanism. A key limitation of AMA is that a bidder’s payment is positively correlated with the bids of others. This is not an issue when valuations are independent, but when valuations are negatively correlated, the paper shows that the performance of AMA can be arbitrarily bad. To address this, the authors propose Correlation-Aware AMA (CA-AMA), which introduces a new payment term for each bidder that depends only on the other bidders' reports. This added term increases the expressiveness of the mechanism but comes at the cost of not always satisfying IR. The authors provide examples with correlated valuations where CA-AMA significantly outperforms AMA. They also present an empirical framework for learning the optimal CA-AMA for a given instance, and conduct numerical experiments to demonstrate the effectiveness of their learning approach.

**Questions:**

It seems that the correlation-aware payment is an arbitrary function of other bidders' valuation. What are the major benefits of such a big freedom? Can you limit correlation-aware payment to a simpler form?

**Ethical Concerns:**

["NO or VERY MINOR ethics concerns only"]

**Final Justification:**

Though this paper lacks strong theoretical insights and the technical contribution feels incremental, it demonstrates practical value in real-world settings. I appreciate the authors’ efforts in addressing the modifications and directions suggested by me and the other reviewers. Overall, I have decided to keep my positive score unchanged (weak accept).

**Limitations:**

yes

**Quality:**

3

**Strengths And Weaknesses:**

Strengths:
1. The paper is well-written and easy to follow.
2. The problem is well-motivated.
3. The numerical experiments are comprehensive.

Weaknesses:

1. The problem looks a bit incremental, it simply adds a payment term to AMA.
2. IMHO, one major downside is that is makes the optimization problem much more complicated by introducing the IR constraint. In the original AMA formulation, IR is not a constraint of optimization since AMAs guarantee IR.
3. The theoretical analyses and guarantees of CA-AMA is not very strong -- it mainly covers the single-bidder case.

---

> ### Author Rebuttal · Authors · 2025-07-31
>
> We thank the reviewer for the insightful feedback and constructive suggestions. We address the specific points below.
>
> ---
> **Weakness 1**: The problem looks a bit incremental, it simply adds a payment term to AMA.
>
> **Response:** We understand the concern that our proposed mechanism might appear to be an incremental addition to the classic AMA by simply incorporating a payment term. However, we respectfully clarify that **this addition is a crucial enhancement that addresses a significant limitation of AMA**, leading to substantial improvements in both theoretical expressiveness and empirical performance.
>
> Our theoretical analysis rigorously demonstrates the importance of this modification. We show that in specific correlated distributions, deterministic AMA can perform arbitrarily poorly compared to our proposed CA-AMA. Furthermore, there's a strict gap in performance between any randomized AMA and CA-AMA. These theoretical findings highlight a fundamental flaw in the classic AMA that our approach rectifies.
>
> Empirical validation further supports our claims. As illustrated in both Figure 2 and Figure 4 of our experiments, CA-AMA consistently shows significant improvement over the classic AMA. This practical superiority, combined with our theoretical guarantees, underscores the non-incremental nature and impactful contribution of our work.
>
> Moreover, **the simplicity of our modification can be a strength, as it's straightforward to implement and integrates seamlessly with existing AMA-based methods.** This means that despite its apparent simplicity, our work offers an efficient and valuable extension to the AMA framework, providing novel insights into optimal multi-bidder, multi-item auction theory.
>
> ---
> **Weakness 2**: It makes the optimization problem much more complicated by introducing the IR constraint.
>
> **Response**: We acknowledge that introducing the IR constraint for regret minimization adds complexity to the optimization problem. However, we firmly believe this extension is justified and offers significant advantages.
>
> **IR-regret is computationally efficient compared to other methods involving computing a regret term.** For instance, computing the IC-regret, which is used in RegretNet and its successors, requires finding an approximate best-bid strategy and computing a large number of additional auctions. The feasibility regret in GemNet involves complex integer programming post-processing. Compared to these, although CA-AMA is still not universally expressive, **its training time is comparable to classic AMA, which is much more efficient (as shown in Table 2).** The hyperparameter for IR-regret minimization is also less sensitive under our updating rule (see equation below Line 213). Furthermore, our theoretical analysis confirms that CA-AMA is strictly more expressive than AMA, effectively addressing its key limitations.
>
> ---
> **Weakness 3**: The theoretical analyses mainly cover the single-bidder case.
>
> **Response**: **We conjecture that the result “CA-AMA = AMA in bidder-independent auctions” (first part of Theorem 3.3) extends to multi-item auctions**. The empirical result can provide some evidence: in bidder-independent cases, $\alpha=0$ in Table 1, the performance of CA-AMA is similar to AMA. We plan to further explore this extension, but this is out of the scope of this paper. We would like to discuss the challenge arising in the multi-item case.
>
> Firstly, the optimal revenue in multi-item settings is generally unknown and remains a fundamental open problem in game theory. Even for AMA, there are no existing results that characterize the optimal AMA in general valuation distributions. This makes it challenging to utilize the properties of the optimal AMA.
>
> Secondly, in multi-item auctions, the allocation of one item can depend on the allocation of others. For example, an item might be reserved if bidders' valuations for other items are low. This introduces significant differences between the single-item and multi-item cases, resulting in more discontinuities in allocations.
>
> We believe further exploration on this topic would help to tackle the fundamental challenge of optimal auction design for a general multi-item auction.
>
> ---
> **Question 1**: It seems that the correlation-aware payment is an arbitrary function of other bidders' valuations. What are the major benefits of such a big freedom? Can you limit correlation-aware payment to a simpler form?
>
> **Response**: The correlation-aware payment is modeled as an arbitrary function to leverage the flexibility of neural networks for practical implementation. As derived in the paper (equation below Line 245), the target form of the correlation-aware payment is:
>
> $$p_i^{\text{OPT-core}}(V_{-i}) = \inf_{v_i \in \text{supp}(\mathcal F_i(V_{-i}))} u_i^{\text{AMA}}((v_i, V_{-i}); \mathcal A, w, \lambda).$$
>
> Then we further compute the utility under the AMA mechanism, $u_i^{\text{AMA}}$. Specifically, for a fixed menu $\mathcal A$ whose size is $K$, the utility is
> $$u_i^{\text{AMA}} = \max\_{k \in [K]} \left(\sum\_{j=1}^N w_j v_j \cdot (A_k)_j + \lambda\_k\right) - \max\_{k \in [K]} \left(\sum\_{j=1, j\neq i}^N w_j v_j \cdot (A_k)_j + \lambda\_k\right).$$
> As shown in the equation, the payment is a minus of two maximums of multiple linear functions. Therefore, in theory, a structure with this property can characterize the target correlation-aware function.
>
> Inspired by this, we conduct additional experiments that use a simpler architecture, MaxMinusMaxNet, defined as:
>
> ```python
> class MaxMinusMaxNet(nn.Module):
>     def __init__(self, input_dim: int, k: int):
>         super().__init__()
>         self.linear_branch1 = nn.Linear(input_dim, k)
>         self.linear_branch2 = nn.Linear(input_dim, k)
>
>     def forward(self, x: torch.Tensor):
>         out1 = self.linear_branch1(x)
>         max1, _ = torch.max(out1, dim=1)
>         out2 = self.linear_branch2(x)
>         max2, _ = torch.max(out2, dim=1)
>         return max1 - max2
> ```
>
> The results demonstrate that this architecture yields results comparable to the MLP, indicating robustness to simpler forms. As the results are similar, we will maintain the experimental results in the main paper and add a discussion about this issue in the final version.

---

> > ### Comment · Reviewer_YyPf · 2025-08-04
> >
> > Thank you for the detailed response! I appreciate the authors' efforts in exploring the modifications and directions suggested by me and other reviewers. I’ve decided to keep my positive score unchanged.

---

### Note · Authors · 2025-08-12

We sincerely thank all reviewers for their constructive feedback and productive discussions during the rebuttal phase. We are particularly encouraged that **all reviewers provided positive scores**. Notably, **Reviewers YyPf and TSDr have confirmed that our rebuttal addressed their concerns**, and we believe our responses also resolve Reviewer Csj4’s points, as their concerns are closely aligned.

This paper introduces *Correlation-Aware Affine Maximizer Auctions* (CA-AMA), which integrates the classic AMA with a correlation-aware payment rule. Our theoretical results demonstrate that AMA can perform arbitrarily poorly compared to CA-AMA in certain cases, and we provide sufficient conditions under which CA-AMA and AMA exhibit equal expressiveness. Extensive experiments, conducted in settings consistent with prior work, confirm that CA-AMA improves revenue over AMA with minimal additional training complexity.

Reviewer concerns primarily focused on three points:

1. **Influence of introducing IR-regret**

   We demonstrate that IR-regret can be efficiently computed and optimized more effectively than IC-regret and feasibility-regret. Additionally, we propose a post-transformation to ensure strict IR. See our detailed responses to *Question* (Reviewer Csj4), *Weakness 2* (Reviewer YyPf), and *Weakness 1* (Reviewer TSDr).

2. **Revenue improvement of CA-AMA over AMA in certain distributions**

   We show that the performance in settings with less correlated valuations is expected, as predicted by our theoretical result demonstrating that CA-AMA has equivalent expressiveness to AMA under specific conditions. See our detailed responses to *Weakness* (Reviewer Csj4) and *Weakness 1* (Reviewer YyPf).

3. **Extension of theoretical results to multi-item auctions**

   We discuss the challenges of extending the AMA vs. CA-AMA comparison to multi-item settings and outline a conjecture and proof idea that we believe will guide future work. See our detailed responses to *Weakness 3* (Reviewer YyPf) and *Weakness 5* (Reviewer TSDr).

We are committed to incorporating all feedback for greater clarity and believe this paper will be a valuable contribution to the NeurIPS community. We again thank the reviewers and the Area Chairs for their time, insights, and constructive engagement.

---

### Decision · Program_Chairs · 2025-09-17

**Decision:**

Reject

**Comment:**

Overall, reviewers are all borderline positive about the paper. However none of them shows strong support/excitement. They think the paper studies a well-motivated problems and has novel ideas. And they have concerns on practical relevance and technical contributions. After reading the reviews and rebuttal, I do share similar concerns.